# Improving the predictions of black carbon (BC) optical properties at various aging stages using a machine-learning-based approach

Baseerat Romshoo[1,2,*], Jaikrishna Patil[3,a,*], Tobias Michels[3], Thomas Müller[1], Marius Kloft[3], and Mira Pöhlker[1,2,4]

[1]Atmospheric Microphysics Department, Leibniz Institute for Tropospheric Research, 04318 Leipzig, Germany
[2]Multiphase Chemistry Department, Max Planck Institute for Chemistry, 55128 Mainz, Germany
[3]Department of Computer Science, RPTU Kaiserslautern-Landau, 67653 Kaiserslautern, Germany
[4]Faculty of Physics and Earth Sciences, Leipzig Institute for Meteorology, Leipzig University, 04103 Leipzig, Germany
[a]now at: Arizona State University, 699 S Mill Ave, Tempe, AZ, 85281, USA
[*]These authors contributed equally to this work.

**Correspondence:** Baseerat Romshoo (baseerat@tropos.de) and Marius Kloft (marius.kloft@cs.rptu.de)

**Abstract.** It is necessary to accurately determine the optical properties of highly absorbing black carbon (BC) aerosols to estimate their climate impact. In the past, there has been hesitation about using realistic fractal morphologies when simulating BC optical properties due to the complexity involved in the simulations and the cost of the computations. In this work, we demonstrate that using a benchmark machine learning algorithm, it is possible to make highly fast and accurate predictions of the optical properties for BC fractal aggregates. The mean absolute errors (MAE) for the optical efficiencies ranged between 0.002 and 0.004, whereas they ranged between 0.003 and 0.004 for the asymmetry parameter. Unlike the computationally intensive simulations of complex scattering models, the ML-based approach accurately predicts optical properties in a fraction of a second. Physiochemical properties of BC, such as total particle size (number of primary particles ($N_{pp}$), outer volume equivalent radius ($r_o$), and mobility diameter ($D_m$), outer primary particle size ($a_o$), fractal dimension ($D_f$), wavelength ($\lambda$), and fraction of coating ($f_{\text{coating}}$) were used as input parameters for the developed ML-algorithm. An extensive evaluation procedure was carried out in this study while training the ML algorithms. The ML-based algorithm compared well with observations from laboratory-generated soot, demonstrating how realistic morphologies of BC can improve their optical properties. Predictions of optical properties like single scattering albedo ($\omega$) and mass absorption cross-section ($MAC$) were improved compared to the conventional Mie-based predictions. The results indicate that it is possible to generate optical properties in the visible spectrum using BC fractal aggregates with any desired physicochemical properties within the range of the training dataset, such as size, morphology, or organic coating. Based on these findings, climate models can improve their radiative forcing estimates using such comprehensive parameterizations for the optical properties of BC based on their aging stages.

## 1 Introduction

Black carbon (BC) aerosols are strong absorbers of solar radiation formed from incomplete combustion of fossil fuels, biofuels, and biomass (Ramanathan and Carmichael, 2008; Bond et al., 2013). In the atmosphere, BC is usually found together with other types of aerosols, which form a coating around it (Sun et al., 2022; Sedlacek III et al., 2022; Romshoo et al., 2023a).

To understand the impact of BC on the environment, global climate models require information about its light scattering and absorption properties (Jacobson, 2001). The most common morphology assumed for such BC-containing aerosols in light scattering codes is a spherical core-shell shape (Bond et al., 2013). The Lorentz-Mie theory (Mie, 1908) is often used to calculate the optical properties of such spherical BC particles (Bohren and Huffman, 2008). However, studies have shown significant discrepancies in the results of Lorentz-Mie theory when compared with ambient measurements (Romshoo et al., 2024; Adachi et al., 2010; Wu et al., 2018).

High-resolution Transmission Electron Microscopy (TEM) images showed that the BC particles have a fractal structure composed of numerous spherules known as primary particles (Chakrabarty et al., 2006). This led to an advanced mathematical description of BC as fractal aggregates, known as fractal law (Mishchenko et al., 2002):

$$N_{pp} = k_f \left( \frac{R_g}{a} \right)^{D_f},  \tag{1}$$

where $a$ is the radius of the primary particle, $N_{pp}$ is the number of primary particles, $k_f$ is the fractal prefactor, and $D_f$ is the fractal dimension. $R_g$ is the radius of gyration, which characterizes the spatial size of the aggregate. The shortcomings of the simplified spherical assumption of BC have caused the scientific community to develop towards the use of such realistic fractal aggregate morphology for computing the optical properties of BC (e.g., Kahnert and Kanngießer, 2020; Romshoo et al., 2021; Kahnert, 2010a; Wu et al., 2018; Liu and Mishchenko, 2018).

Romshoo et al. (2022) showed that the discrepancy between modeled and measured optical properties could be reduced to 10% when an aggregate morphology is used. To simulate the optical properties of BC as fractal aggregates, the most commonly used methods are the Rayleigh-Debye-Gans (RDG) approximation (Sorensen, 2001), the discrete dipole approximation DDA (Purcell and Pennypacker, 1973), the Generalized Multi-particle Mie (GMM) method (Xu and Gustafson, 2001) and the T-matrix method (Mishchenko et al., 1996). The Multi Sphere T-Matrix (MSTM) method has found widespread applications in the research field because of its high computational speed and accuracy in comparison to other methods like the DDA (Kahnert and Kanngießer, 2020; Yurkin and Kahnert, 2013). Although MSTM has lower computational costs when compared to other numerical methods, a single simulation can still take more than 24 hours, depending on the properties of the aggregate.

Consequently, pre-calculated databases have been developed for aggregate properties to save time for constructing detailed aggregates and the time-consuming optical simulations (Liu et al., 2019; Romshoo et al., 2021). Using these databases as look-up tables mitigates high computational overhead in large-scale applications. Still, this approach is limited by the range and step size of parameters chosen during the database creation. Previous work has trained machine learning (ML) models on such databases (Luo et al., 2018a; Lamb and Gentine, 2023) to overcome those limitations. Once trained, those ML models provide predictions for BC optical properties in a fraction of a second. Luo et al. (2018a) train a support vector regressor on a database generated using MSTM simulations ($N_{pp}$ from 8 to 3000; $D_f$ from 1.8 to 2.2). However, they did not consider coating and used pure BC aggregates in their experiments. Their results also suggest that their model has considerable difficulties when attempting to predict optical properties for physicochemical properties not in the range of the training data. Lamb and Gentine (2023) predicted optical properties of uncoated BC fractal aggregate using a graph neural network ($N_{pp}$ from 8 to 960; $D_f$ from 1.8 to 2.3). The input graph contains one node for each primary particle and an edge between two nodes if the distance

between the corresponding primary particles is less than some threshold. The authors generate their ground truth database using the MSTM algorithm, but, like Luo et al. (2018a), they do not consider any coating in their experiments. The machine learning methods, training parameters, performance metrics, and other details of Luo et al. (2018a) and Lamb and Gentine (2023) are compared to this study in Table B1.

This study demonstrates the use of a machine-learning-based approach to predict the optical properties of BC aggregates at various aging stages, including coating, which is highly relevant for atmospheric aerosols. Combining this ML-based approach with a laboratory dataset showed optical properties like single scattering albedo ($\omega$) and mass absorption cross-section ($MAC$) can be predicted more accurately than with conventional Mie-based methods. A database of optical and physicochemical properties of BC has been built for this study, which is an extension to the previous work by Romshoo et al. (2021). We

trained two ML methods on this database: kernel ridge regression (KRR) and artificial neural networks (ANN). Experiments show that these models predict the optical properties of BC aggregates regardless of their size, morphology, or composition at low computational costs and with high accuracy. The dataset used to train our ML models is freely available at Zenodo [1]. Furthermore, we publish our ML models at GitHub[2] together with an easy-to-use wrapper script to allow for integration into higher-level applications. Our approach contributes to improving global climate model radiative forcing estimates by

parametrizing BC optical properties using realistic fractal aggregate morphology.

    The manuscript is structured as follows: Section 2 provides an overview of the physical, chemical, and optical properties of BC used in this study. Section 3 describes the machine learning techniques, including the data processing, machine learning algorithms, and evaluation procedures. In Section 4, the results demonstrate that realistic morphologies of BC can be used to accurately predict optical properties at various stages of aging. Section 5 discusses how the results compare to laboratory

measurements of BC, discussing the atmospheric processing in detail. Potential limitations and challenges of this work in presented in Section 6, ending with the main conclusions in Section 7.

## 2   Database of physicochemical and optical properties of black carbon fractal aggregates

The database for the physicochemical and optical properties of BC fractal aggregates has been designed to consider all the possible aging stages of BC. The optical properties of BC fractal aggregates are most sensitive to the change in particle size

as they age (Matsui et al., 2018). The particle size is reported as dependent parameters of the number of primary particles ($N_{pp}$), volume equivalent radii ($r_i$ and $r_o$), and mobility diameter ($D_m$). Further, the chemical composition and morphology also influence their optical properties. There are constants related to the particle's chemical composition, such as density and refractive index. The optical properties have been reported as efficiencies and cross-sections. Further dependent optical properties have also been included. The mass and volume of the BC particles were used for conversion between various optical

parameters. Further, some parameters, such as the wavelength, were related to the optical model. The database was created using 6192 particles of varying sizes, morphologies, and coating fractions. There are 35 features in the database, which are

---

[1]https://zenodo.org/records/7523058

[2]https://github.com/jaikrishnap/Machine-learning-for-prediction-of-BCFAs

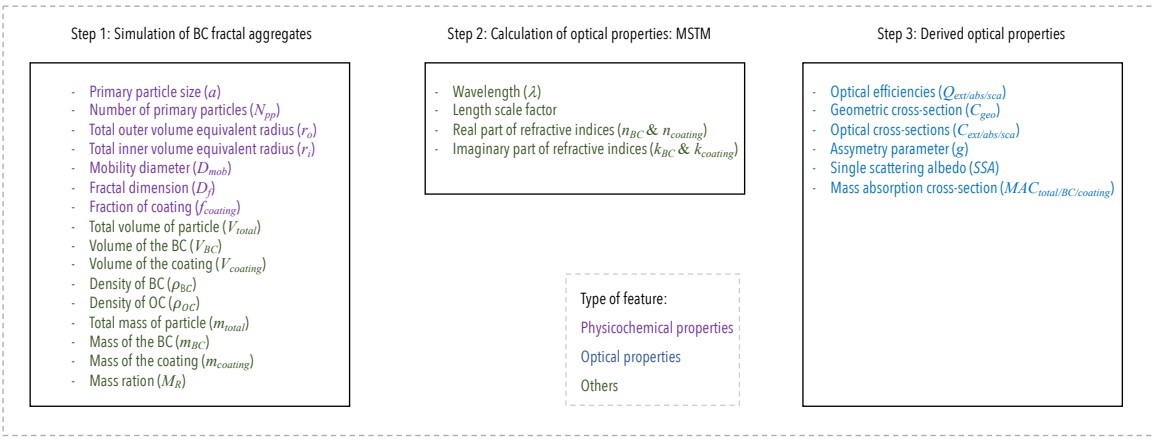

**Figure 1.** Overview of the various features of the database for physicochemical and optical properties of black carbon fractal aggregates. The features are arranged based on the three steps of constructing this database. As the legend at the bottom indicates, the features are further divided into physicochemical properties, optical properties, and others.

categorized into 15 physicochemical features, 13 optical features, and seven constants. In Fig. 1, you can see an overview of all the features of the database. In Table A1, the upper and lower bounds of the main features are provided.

### 2.1 Physicochemical features of the database

The BC fractal aggregate's physicochemical features include size, mass, volume, morphology, and composition. Figure 2 gives some examples of the various BC aggregate particles generated in this study. All the relevant properties provided in the study are discussed below, and their formulas are given in Appendix A1.

#### 2.1.1 Size

*Primary particle size* ($a$). The primary particle size of a BC fractal aggregate is sensitive to the emission source or flame condition. Biomass burning produces black carbon aggregates with comparatively large primary particles, ranging from 15 to 25 nm in radius (Chakrabarty et al., 2006). Diesel engines produce aggregates whose primary particle radius ranges between 10 and 12 nm (Guarieiro et al., 2017). On the other hand, emissions from aircraft engines consist of particles with a radius as small as 5 nm (Liati et al., 2014). There has also been research indicating that the size distribution of primary particles is largely polydisperse (Bescond et al., 2014). Liu et al. (2015) pointed out that when considering a monodisperse and polydisperse distribution of the radius of the primary particle, their resultant radiative properties differ. However, Kahnert (2010b) has shown that particle light absorption is insensitive to the radius of primary particles when they are between 10 nm and 25 nm.

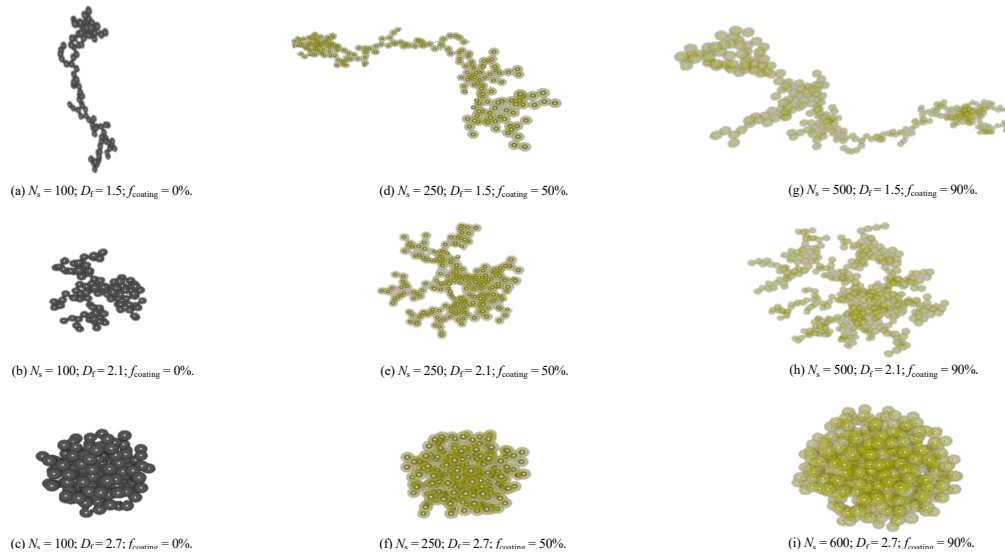

(a) $N_s = 100$; $D_f = 1.5$; $f_{coating} = 0\%$.     (d) $N_s = 250$; $D_f = 1.5$; $f_{coating} = 50\%$.     (g) $N_s = 500$; $D_f = 1.5$; $f_{coating} = 90\%$.

(b) $N_s = 100$; $D_f = 2.1$; $f_{coating} = 0\%$.     (e) $N_s = 250$; $D_f = 2.1$; $f_{coating} = 50\%$.     (h) $N_s = 500$; $D_f = 2.1$; $f_{coating} = 90\%$.

(c) $N_s = 100$; $D_f = 2.7$; $f_{coating} = 0\%$.     (f) $N_s = 250$; $D_f = 2.7$; $f_{coating} = 50\%$.     (i) $N_s = 600$; $D_f = 2.7$; $f_{coating} = 90\%$.

**Figure 2.** Visualization of the various BC aggregate particles generated in this study. Fresh BC aggregates with no external coating are shown in (a) to (c). Semi-aged BC aggregates with 50% coating are shown in (d) to (f). Aged BC aggregates with 90% coating are shown in (g) to (i)

The black carbon fractal aggregates in this study have a monodisperse distribution of the radius of the primary particle. BC aggregates were simulated with the inner diameter of the primary particle ($a_i$) fixed at 15 nm. In contrast, the outer radius of the primary particle ($a_o$), consisting of the organics, varied between 15.1 nm to 30 nm with the fraction of coating ($f_{coating}$ following Eq. (A3) in the Appendix section. The $a_o$ was 15, 15.1, 15.3, 15.5, 15.8, 16.2, 16.5, 16.9, 17.8, 18.9, 20.4, 22.4, 25.6, and 29 according to the value the $f_{coating}$ given in Table A1.

*Number of primary particles ($N_{pp}$).* The number of primary particles determines the overall size of the particle. The BC fractal aggregates were simulated by varying $N_{pp}$ by 5%, starting from 1 up to 1000.

*Volume equivalent radius ($r$).* The volume equivalent radius is defined as the radius of a sphere having the same volume as the BC fractal aggregate, described in Eq. (A1) in the Appendix section. The outer volume equivalent radius ($r_o$) was calculated for the whole BC aggregate and the coating using $a_o$. The inner volume equivalent radius ($r_i$) was calculated using $a_i$ for the BC aggregate without the coating, i.e., pure BC.

*Mobility diameter ($D_m$).* The mobility diameter is the diameter of a sphere with the same migration velocity in a constant electric field as that of the BC fractal aggregate (Flagan, 2001). Mobility size spectrometers can measure $D_m$, which is interesting for ambient and laboratory studies. We derived $D_m$ for the entire range of $N_{pp}$ using the conversion given by Sorensen (2011)—see Eq. (A2) in the Appendix section.

*Geometric cross-section ($C_{geo}$).* The geometric cross-section is the area of the cross-section of a volume-equivalent sphere given as Eq. (A4) in the Appendix section.

### 2.1.2 Mixing state

Along with BC, a complex mixture of gas-phase organic compounds is co-emitted during incomplete combustion, forming a coating around the BC aggregates (Gentner et al., 2017). As the BC aggregates stay in the atmosphere, they transform from being hydrophobic to hydrophilic due to water deposition attracting other foreign coatings (Bhandari et al., 2019). The result is that BC particles undergo complex changes in their morphology throughout atmospheric aging, transforming from bare to partially coated aggregates and finally forming compact spherical structures embedded within external coatings (Coz and Leck, 2011; Joel C. Corbin and Gysel-Beer, 2023). Therefore, considering BC as fractal aggregates is necessary to represent all the different stages during their atmospheric aging process. The two parameters describing the mixing state are:

*Fractal dimension* ($D_f$). The fractal dimension is a parameter for morphology that quantifies the folding of BC fractal aggregates into spherical structures with increasing residence time. The value of $D_f$ increases as an aggregate grows into a more spherical frame. A $D_f$ of 3 is the maximum value describing a complete sphere, whereas $D_f$ of 1 represents an early stage open-chain-like aggregate. In the early stages of the BC aging cycle, $D_f$ is usually between 1.5 and 1.9 (Wentzel et al., 2003). With increasing residence time in the atmosphere, aggregates become more compact with a fractal dimension of up to 2.2 (Wang et al., 2017). A humid environment or foreign coatings may further reshape the BC fractal aggregates into more compact structures with a fractal dimension of up to 2.6 (Bambha et al., 2013). In this study, the range of fractal dimensions was taken from 1.5 to 2.9 with a step size of 0.2.

*Fraction of coating* ($f_\text{coating}$). The fraction of coating is the percentage of coating volume compared to the total volume of the BC fractal aggregate. To cover all aging stages, the coating fraction was taken from 1% to 90% in increments of 5%. Note that the coating composition was constrained to non-absorbing organics in this study. $f_\text{coating}$ is dependent on the $a_o$ and $a_i$, described by Eq. (A3) in the Appendix section.

### 2.1.3 Others

*Volume*. Three features in our database describe the volume of a BC aggregate: 1. Total volume of the particle ($V_\text{total}$), 2. the volume of the BC ($V_\text{BC}$), and 3. the volume of the organic coating ($V_\text{coating}$).

*Mass*. Similarly, we include five features related to the mass of the BC aggregate: 1. Total mass of the particle ($m_\text{total}$), 2. the mass of the BC ($m_\text{BC}$), 3. the mass of the coating ($m_\text{coating}$), 4. the mass ratio of total mass to BC mass $\left(\frac{m_\text{total}}{m_\text{BC}}\right)$, and 5. the mass ratio of coating mass to BC mass $\left(\frac{m_\text{coating}}{m_\text{BC}}\right)$. We computed those values fixing the density of BC as $\rho_\text{BC} = 1.8 \text{gcm}^{-3}$ (Park et al., 2004), and the density of the organic coating as $\rho_\text{OC} = 1.1 \text{gcm}^{-3}$ (Schkolnik et al., 2007).

*Wavelength ($\lambda$)*. The optical properties were calculated in the visible spectrum, i.e., for $\lambda \in \{467\text{nm}, 530\text{nm}, 660\text{nm}\}$.

## 2.2 Optical model and the optical features of the database

The tunable Diffusion Limited Aggregation (DLA) software (Wozniak et al., 2012) was used to simulate bare BC fractal aggregates of various physicochemical properties. BC can exhibit a range of coating thicknesses and fractal dimensions at any point in the atmosphere, as evidenced by images from Transmission Electron Microscopy (TEM) analyzed from different

locations (Fu et al., 2012). Detailed information and images from TEM analysis of BC particles have been provided in the Supplementary. The coating model used in this study is called the "closed-cell model," the results showed good comparability with the realistic coating model (Kahnert, 2017). The MSTM calculates the electromagnetic properties of a system that consists of a set of spheres (Mishchenko et al., 2004; Mackowski and Mishchenko, 2011). In this study, we use MSTM version 3.0

(Mackowski, 2013) written in FORTRAN to compute the electromagnetic properties for fixed and random orientations. For every BC fractal aggregate, the MSTM algorithm presents an orientational average of the combined spherical expansions of each primary particle. The MSTM code is best suited to calculate the optical properties of coated BC fractal aggregates since it consists of nested spheres. However, a limiting condition in MSTM is that primary particles cannot overlap.It was necessary to use this closed-cell coating model due to the non-overlapping sphere limitation of the MSTM code. A sophisticated coating

model would be a good choice, but it requires more complex scattering models, such as Discrete Dipole Approximation (DDA), which is computationally expensive. The optical features of the database are given below:

The real ($n$) and imaginary ($k$) part of the refractive indices for BC and coating (non-absorbing organics) at different wavelengths (Kim et al., 2015) used in this study are summarized in Table A2.

*Optical efficiencies ($Q_{ext/abs/sca}$)*. The MSTM directly calculates the extinction efficiency ($Q_{ext}$), absorption efficiency ($Q_{abs}$),

and scattering efficiency ($Q_{sca}$) of the BC aggregate.

*Optical cross-sections ($C_{ext/abs/sca}$)*. The optical cross-section is the product of efficiency and geometric cross-section—see Eq. (A5) in the Appendix section.

*Asymmetry parameter ($g$)*. The asymmetry parameter is directly obtained from the MSTM, defined as the intensity-weighted average of the cosine of the scattering angle (Eq. (A6) in the Appendix section).

*Single scattering albedo ($\omega$)*. The single scattering albedo is the ratio of scattering efficiency ($Q_{sca}$) and extinction efficiency ($Q_{ext}$), given as Eq. (A7) in the Appendix section.

*Mass absorption cross-section (MAC)*. The mass absorption cross section is calculated from the ratio of absorption cross section ($C_{abs}$) and mass ($m$) as detailed in Eq. (A8) in the Appendix section. The three kinds of MAC calculated in this study are total mass absorption cross-section ($MAC_{total}$), BC mass absorption cross-section ($MAC_{BC}$), and coating mass absorption

cross-section ($MAC_{coating}$).

## 3   Machine learning method for predicting optical properties of BC fractal aggregates

As mentioned in Section 1, several high-impact applications, such as climate modeling (Jacobson, 2001), depend on accurate optical properties for specific BC particles. Hence, we propose to train an ML model on a pre-computed database containing physicochemical and corresponding optical properties of BC fractal aggregates at several life cycle stages. This model will learn

patterns and structures within the data and should generalize to unseen data values when used in applications, as evidenced by the success of ML in several domains (Radford et al., 2021; Ramesh et al., 2022). In this work, we train kernel ridge regression and a multi-layer perceptron on the database introduced in Section 2. The following sections detail our data processing routines, models, and evaluation procedures.

## 3.1 Data pre-processing

The subset of the database used as input was designed to include the critical parameters that influence the BC optical properties. As mentioned in Section 2.1, not all physical properties in the database are independent, as some can be derived from others using simple formulae. Including all properties as inputs for the ML model will thus present it with redundant information, increasing its computational overhead and possibly even harming its performance. The first criterion to narrow down the input parameters was broadly choosing the independent physicochemical parameters representing particle size and mixing state.

The fractal dimension ($D_f$) was used to represent the morphology of the BC fractal particles. The chemical mixing state is represented by the fraction of coating ($f_{\text{coating}}$). The Wavelength ($\lambda$) is also an input parameter. There was an exception in selecting the input parameters for particle size where we decided to keep four dependent parameters of outer primary particle size ($a_o$), number of primary particles ($N_{pp}$), outer volume equivalent radii ($r_o$), and mobility diameter ($D_m$). The reason for including all four size parameters is that depending upon the focus of a study, the user may have one or another parameter representing the size. In this way, we could provide a more user-friendly prediction script in which the user has a choice to enter

representing the size. In this way, we could provide a more user-friendly prediction script in which the user has a choice to enter at least one or more of the four size parameters. Therefore, the subset of the database's properties as input for our ML models is $\lambda$, $D_f$, $f_{\text{coating}}$, $a_o$, $N_{pp}$, and $r_o$, and $D_m$. The range of each input parameter used for designing the prediction algorithm is summarised in Table A1. The selection of input parameters needed while running the prediction script: $\lambda$, $D_f$, $f_{\text{coating}}$, and at least one among the $N_{pp}$, and $r_o$, and $D_m$.

Similarly, a BC fractal aggregate's optical properties are also not independent. Thus, we make the ML model predict only the following three properties and compute the rest using the formulae in Appendix A1: absorption efficiency ($Q_{\text{abs}}$), scattering efficiency ($Q_{\text{sca}}$), and asymmetry parameter ($g$).

After feature selection, we transform input features using the Box-Cox transformation (Box and Cox, 1964), where we choose the transformation parameter by maximum-likelihood estimation. We also tried to apply the Box-Cox transformation

to the target features, but since this did not improve results, we decided not to use any transformation on the target features for the experiments that we report in Section 4. To find a suitable regression model, we conducted experiments with multiple ML-based models for regression, including Support Vector Regression (SVR), Ridge Regression (RR), Kernel Ridge Regression (KRR), and Artificial Neural Network (ANN). Each model was evaluated using Mean Absolute Error (MAE) on the sample dataset. The results showed that Kernel ridge regression and Neural Networks demonstrated better performance, especially in

capturing the non-linear relationships within the dataset. Hence, we used KRR and Neural Networks for further analysis.

## 3.2 Kernel ridge regression

Given a labeled dataset of $N \in \mathbb{N}$ points $\{(\boldsymbol{x}^{(1)}, \boldsymbol{y}^{(1)}), (\boldsymbol{x}^{(2)}, \boldsymbol{y}^{(2)}), \ldots, (\boldsymbol{x}^{(N)}, \boldsymbol{y}^{(N)})\} \subset \mathbb{R}^D \times \mathbb{R}^{D'}$, the *regression* problem consists of finding a function $f \colon \mathbb{R}^D \to \mathbb{R}^{D'}$ such that $f(\boldsymbol{x}^{(n)}) \approx \boldsymbol{y}^{(n)}$ for all $n \in \{1, \ldots, N\}$. *Kernel ridge regression* (KRR) (Shawe-Taylor and Cristianini, 2004) learns a function of the form $f(\boldsymbol{x})_d = \sum_{n=1}^N \alpha_{nd}^* k(\boldsymbol{x}^{(n)}, \boldsymbol{x})$, where $k \colon \mathbb{R}^D \times \mathbb{R}^D \to \mathbb{R}$

is a positive semidefinite *kernel function* (Cortes and Vapnik, 1995) and $\alpha^* \in \mathbb{R}^{N \times D'}$ is a solution of the following convex

optimization problem:

$$\min_{\boldsymbol{\alpha} \in \mathbb{R}^{N \times D'}} \lambda \operatorname{Tr}(\boldsymbol{\alpha}^T \mathbf{K} \boldsymbol{\alpha}) + \|\mathbf{Y}^T - \mathbf{K}\boldsymbol{\alpha}\|_{\text{Fro}}^2, \qquad (2)$$

where $\mathbf{K} \in \mathbb{R}^{N \times N}$ is the so-called *kernel matrix* defined by $K_{ij} = k(\boldsymbol{x}^{(i)}, \boldsymbol{x}^{(j)})$, $\lambda \in \mathbb{R}^+$ is a trade-off parameter that controls the influence of the regularization term, $\mathbf{Y} = (\boldsymbol{y}^{(1)}, \ldots, \boldsymbol{y}^{(N)})^T \in \mathbb{R}^{N \times D'}$, and $\|\mathbf{Z}\|_{\text{Fro}} := \sqrt{\sum_{n=1}^{N} \sum_{d=1}^{D'} |z_{nd}|^2}$ denotes the Frobenius norm. Note that Eq. (2) has a closed-form solution:

$$\boldsymbol{\alpha}^* := (\mathbf{K} + \lambda \mathbf{I}_N)^{-1} \mathbf{Y}. \qquad (3)$$

A popular choice for the kernel function is the Gaussian or radial basis function (RBF) kernel

$$k(\boldsymbol{x}, \boldsymbol{x}') = \exp\left(-\gamma \|\boldsymbol{x} - \boldsymbol{x}'\|_2^2\right), \qquad (4)$$

where $\gamma \in \mathbb{R}^+$ is a parameter called *bandwidth* and $\|\boldsymbol{x}\|_2 := \sqrt{\sum_{d=1}^{D} |x_d|^2}$ denotes the $L_2$-norm.

We use *scikit-learn*'s KRR implementation[3] with the RBF kernel for our experiments. This method has two hyperparameters that need tuning: the RBF kernel's $\gamma \in \mathbb{R}^+$ and $\lambda \in \mathbb{R}^+$ (see Eq. (2)). We optimize hyperparameters using grid search—please see Table B2 for the grid and Section 3.4 for more detailed information on our evaluation procedure.

## 3.3 Artificial neural networks

*Artificial neural networks* (ANN) constitute one of the founding pillars of ML's success during the last ten years. Originally, their design was inspired by the structure of neurons inside the nervous system of several organisms (Rosenblatt, 1958). Most designs used in practice nowadays abandoned that idea, but the name remains.

In our experiments, we use a *feed-forward* ANN, sometimes also called *multi-layer perceptron* (MLP). It consists of an arbitrary number ($L \geq 2$) of *layers*, of which the first is called *input* layer, the last is called *output* layer, and all layers in between are called *hidden* layers. Each layer consists of a certain number of *neurons*, which are connected to the neurons in the previous and the following layer

Formally, we can define an MLP as a function $f : \mathbb{R}^D \to \mathbb{R}^{D'}$ that is composed of $L - 1$ layer functions, i.e., $f(\boldsymbol{x}) := f^{(L-1)}(f^{(L-2)}(\ldots f^{(1)}(\boldsymbol{x})\ldots))$, where each $f^{(l)} \colon \mathbb{R}^{D^{(l)}} \to \mathbb{R}^{D^{(l+1)}}$ represents a connection between two layers. They are defined as $f^{(l)}(\boldsymbol{x}) := \sigma^{(l)}(\mathbf{W}^{(l)}\boldsymbol{x} + \boldsymbol{b}^{(l)})$, where $\mathbf{W}^{(l)} \in \mathbb{R}^{D^{(l+1)} \times D^{(l)}}, \boldsymbol{b}^{(l)} \in \mathbb{R}^{D^{(l+1)}}$ are learnable parameters, and $\sigma^{(l)}$ is a so-called *activation function* that is applied separately to each element of its input vector. Common choices for $\sigma^{(l)}$ include, for example, the rectified linear unit (ReLU) $\sigma^{(l)}(x) = \max(x, 0)$ or the $\tanh$ function. We use the same activation function for each layer except the last, where we always use the identity function, i.e., $\sigma^{(L-1)}(x) := x$. Finally, $D^{(l)} \in \mathbb{N}$ denotes the number of neurons in layer $l$, with $D^{(1)} = D$ and $D^{(L)} = D'$.

The number of hidden layers, the number of neurons in those hidden layers, and the activation function are usually chosen by a human before training a neural network. Together, they define the *architecture* of the MLP. We can *learn* values for the

---

[3]https://scikit-learn.org/stable/modules/generated/sklearn.kernel_ridge.KernelRidge.html

parameters $\mathbf{W} := (\mathbf{W}^{(1)}, \ldots, \mathbf{W}^{(L-1)})$ and $\boldsymbol{b} := (\boldsymbol{b}^{(1)}, \ldots, \boldsymbol{b}^{(L-1)})$ by minimizing a so-called *loss function* $\mathcal{L} : \mathbb{R}^{D'} \times \mathbb{R}^{D'} \to \mathbb{R}$ over a dataset:

$$\min_{\mathbf{W}, \boldsymbol{b}} \frac{1}{N} \sum_{n=1}^{N} \mathcal{L}(f(\boldsymbol{x}^{(n)}), \boldsymbol{y}^{(n)}). \tag{5}$$

When solving a regression problem, the most common choice for $\mathcal{L}$ is the *squared loss* $\mathcal{L}(\hat{\boldsymbol{y}}, \boldsymbol{y}) := \|\boldsymbol{y} - \hat{\boldsymbol{y}}\|_2^2$, but practitioners sometimes use other loss functions as well, for example, the *Huber loss* (Huber, 1964):

$$\mathcal{L}(\hat{\boldsymbol{y}}, \boldsymbol{y}) = \sum_{d=1}^{D'} \begin{cases} \frac{1}{2}(y_d - \hat{y}_d)^2 & \text{if } |y_d - \hat{y}_d| \leq \delta \\ \delta\left(|y_d - \hat{y}_d| - \frac{1}{2}\delta\right), & \text{otherwise,} \end{cases} \tag{6}$$

where $\delta \in \mathbb{R}^+$ determines the cut-off point between squared and absolute loss and is usually chosen as $\delta = 1$. The entire procedure of adapting the ANN's parameters using a given dataset is called *training* in the ANN literature.

Note that, in general, Eq. (5) is not convex and does not have a closed-form solution. Hence, practitioners use gradient-based optimization methods, i.e., variants of mini-batch stochastic gradient descent (SGD) (Bottou et al., 2018), to find a local
minimum of Eq. (5).

For our experiments, we implemented an MLP using *keras*[4]. Table B3 contains the hyperparameter grid for the MLP's architecture and training procedure.

### 3.4   Evaluation procedure

In the case of kernel ridge regression, regularization is carried out by the regularization constant $\lambda$ (chosen optimal value =
0.0001). For neural networks, we tested the dropout technique to prevent overfitting. However, dropout regularization did not show notable improvements in the model's generalization. After preprocessing, we split the database into a training and a test set. Models perform their training procedures and hyperparameter tuning on the training set only, and we then evaluate the model's performance exclusively on the test set. We consider three different methods of performing this split—each one intends to measure another aspect of the model's performance:

1. *Random split*: We randomly assign each point in the database to either the training or test set. Note that we use 30% of the data for the test set and the rest for the training set. Using this split, the training and test set's feature distribution should be similar. Thus, measuring the performance on the test set produces a general measure of the model's capability to learn the underlying patterns in the data.

2. *Interpolation split*: Here, we choose a feature and a certain range in the middle of that feature's range and choose all data
points within that range as the test set. To achieve high test scores, the model must, therefore be capable to *interpolate*

---

[4]https://keras.io/
[2]https://keras.io/api/layers/activations/#sigmoid-function
[3]http://www.cs.toronto.edu/~tijmen/csc321/slides/lecture_slides_lec6.pdf
[4]https://keras.io/api/losses/regression_losses/#logcosh-class

predictions for data points it has not seen during training. Table B4 shows the features and ranges used for the two interpolation splits. The split was tested for $D_f$ using the training data of $D_f = [1.5, 2.1] \cup (2.5, 2.9]$, whereas, a training data of $[0, 35) \cup (50, 90]$ was used for testing $f_{\text{coating}}$.

3. *Extrapolation split*: Similar to the interpolation splits, we also consider choosing a test set at the boundaries of certain features. This measures the model's extrapolation capabilities. Table B5 shows the features and ranges used for the four different extrapolation splits. The two splits for testing $D_f$ used training data $[1.5, 2.5)$ and $(1.9, 2.9]$. The other two splits for $f_{\text{coating}}$ used training data of $[0, 75)$ and $(15, 90]$.

We use the *mean absolute error* (MAE) as our primary performance metric: given a dataset $\mathcal{D} \subset \mathbb{R}^D \times \mathbb{R}^{D'}$, and our prediction model $f \colon \mathbb{R}^D \to \mathbb{R}^{D'}$ we can compute the MAE as follows:

$$\text{MAE}(f, \mathcal{D}) = \frac{1}{|\mathcal{D}|} \sum_{(\boldsymbol{x}, \boldsymbol{y}) \in \mathcal{D}} \|\boldsymbol{y} - f(\boldsymbol{x})\|_1 \,, \tag{7}$$

where $\|\boldsymbol{z}\|_1 := \sum_{d=1}^{D'} |z_d|$ is the $L_1$-norm.

Regardless of the split strategy, we split the training set once more into a train and a validation set using the random split method during the training phase. Here, we use again 30% of the data for validation and the remaining 70% for training. Our models then train on the train set for all possible hyperparameter configurations defined in the grid, and we record the MAE on the validation set for each combination. Finally, we choose the combination with the lowest MAE and evaluate the corresponding model's MAE on the test set.

## 4   Performance of the machine learning models

The error distributions for the ML methods are presented in Fig. 3, for different experimental scenarios of the data splitting with respect to the parameter fractal dimension. The median error is close to zero for the random and interpolation splits, meaning our models do not generally over- or underestimate any optical value. The distribution of errors (excluding outliers) for the random and interpolation splits is relatively narrow, indicating that most test points have minor errors. In the extrapolation case, both ML models exhibit bias, such as overestimation of $Q_{\text{sca}}$ by the ANN and overestimation of $g$ by the KRR. However, the mean absolute error, even for the extrapolation split, is 1.5 to 8 %, which is still within reasonable limits. Luo et al. (2018a) showed that their model has considerable difficulties when attempting to predict optical properties for parameters not in the range of the training data. However, adding a few data points to extend any parameter range significantly improved the prediction ability of the ML algorithm. The interpolation and extrapolation results are similar if training and test data are split according to the parameters of the coating $f_{\text{coating}}$ and particle size $D_{\text{m}}$. The appendix provides a more detailed discussion about the interpolation and extrapolation results for parameters of $f_{\text{coating}}$ and $D_{\text{m}}$ in Fig. C1 and Fig. C2, respectively. Overall, the narrow box plots of the errors in the random split demonstrate the effectiveness of the ML algorithms for predicting the optical properties of coated BC fractal aggregates.

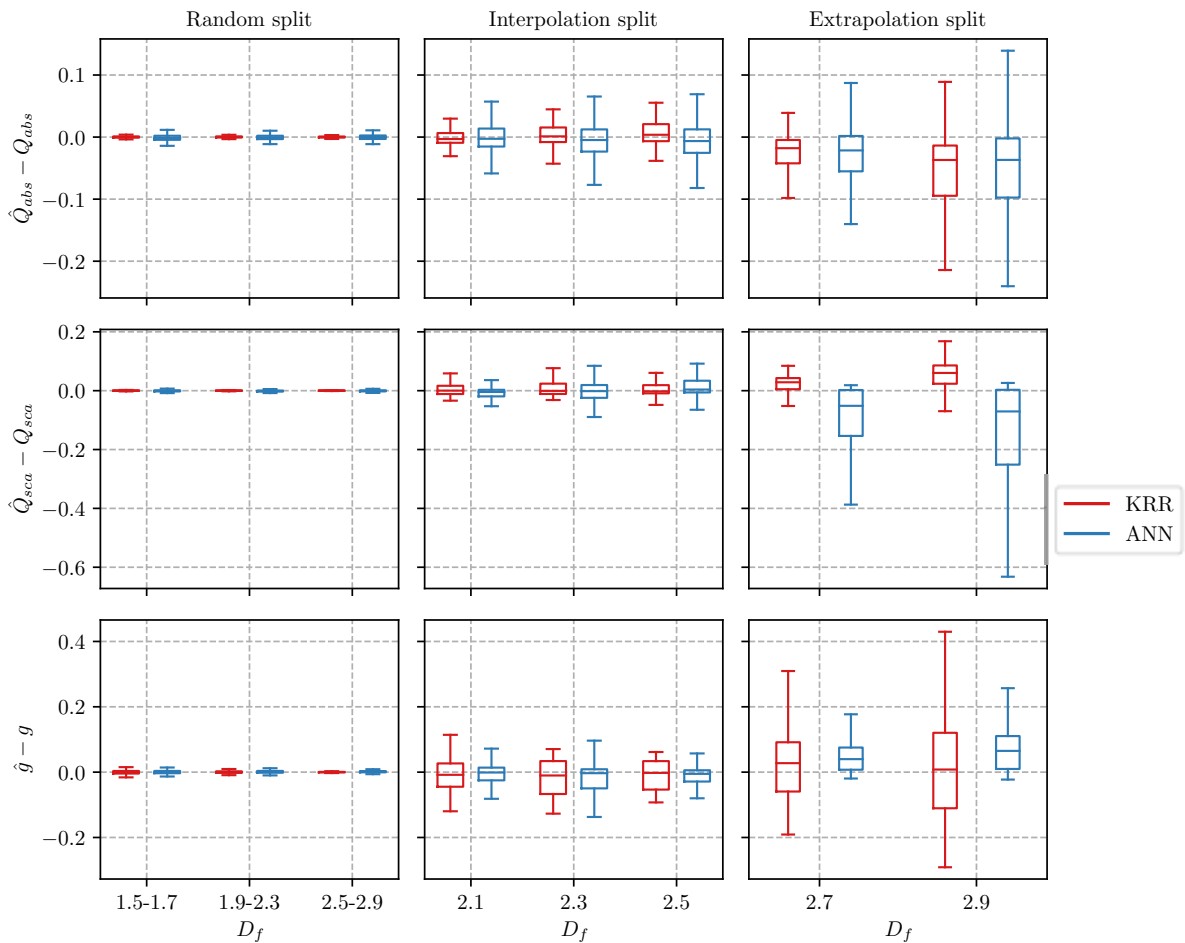

**Figure 3.** Boxplots summarizing the error between the predicted value ($\hat{Q}_{abs}, \hat{Q}_{sca}, \hat{g}$) and the true value for three optical properties. The training data for the interpolation split consists of fractal dimensions in $D_f = [1.5, 2.1] \cup (2.5, 2.9]$, whereas the extrapolation split uses $D_f = [1.5, 2.5]$. The lower hinge and the upper hinge of the boxplot represent the 25 % and 75 % quantile of the observations, respectively. Note that the outliers significantly reduced the visualization of the boxplots and, therefore, were omitted from the figures. However, please all the outliers are considered in the training data and error evaluation.

The MAEs for our experiments are reported in Table 1. In the case of the random split, both ML models are pretty accurate, with the percentage of MAEs ranging from 0.1 to 0.4 % when compared to the average feature range. Lamb and Gentine (2023) reported mean absolute percentage errors (MAPE) between 2 to 9% for their optical predictions, whereas Luo et al. (2018a) reported relative errors between 1 to 5 %. The MAPEs are biased to the magnitude of the true value in the denominator. The same MAE can result in significantly different MAPE depending on the magnitude of true value they are divided with. In our view, the prediction error should be weighted equally for both points, and therefore, we chose the MAE as our error metric. Lamb and Gentine (2023) have also discussed how the bias of MAPEs resulted in higher values of nearly 70% for smaller

particles. Error distributions for the ML methods shown in Fig. 3 are presented in terms of MAPE in the Supplementary.

The comparison of the two ML methods for random split in Table 1 showed that KRR generally results in a lower MAE for

predictions of $Q_{abs}$ and $Q_{sca}$. Contrary to this, the ANN could predict $g$ with a lower MAE. In line with expectations, the MAE for the splits based on interpolation and extrapolation is somewhat higher. The errors, however, are still considered relatively minor compared to the features' range. The extrapolation and interpolation experiments were used to test the performance of the ML algorithm under various scenarios of data available for training. The ML models we publish for use in applications were trained on the entire dataset using the best parameters from the random split experiments. As a result, the errors should

be similar to those we report for the random split here.

**Table 1.** Mean absolute errors of the predicted optical properties for different experiments. The training data for the interpolation split consists of fractal dimensions in $D_f = [1.5, 2.1) \cup (2.5, 2.9]$, whereas the extrapolation split uses $D_f = [1.5, 2.5]$.

| Optical property | Random split | | Interpolation split | | Extrapolation split | | Feature range |
|---|---|---|---|---|---|---|---|
| | KRR | ANN | KRR | ANN | KRR | ANN | |
| $Q_{abs}$ | 0.0022 | 0.0039 | 0.0122 | 0.0287 | 0.0329 | 0.0354 | $0 - 2$ |
| $Q_{sca}$ | 0.0019 | 0.0031 | 0.0224 | 0.0466 | 0.0393 | 0.0939 | $0 - 2$ |
| $g$ | 0.0044 | 0.0038 | 0.0429 | 0.0289 | 0.0879 | 0.0485 | $0 - 1$ |

A one-to-one comparison was performed between the estimates and true values to understand better how the ML methods predict optical properties. Fig. 4 compares the estimated and true values for the wavelength of 660 nm when the training and test data are randomly split. The values of $\hat{Q}_{abs}, \hat{Q}_{sca}, \hat{g}$ obtained from the KRR and ANN methods are compared to the true values derived from the MSTM method. The performance of both ML methods was studied for BC fractal aggregates with

three representative morphologies, and coating fractions ($D_f = 1.5$ & $f_{\text{coating}} = 0\%$; $D_f = 2.1$ & $f_{\text{coating}} = 50\%$; $D_f = 2.7$ & $f_{\text{coating}} = 90\%$). There was reasonable agreement between the KRR and the ANN for all the sub-cases. Therefore, the machine learning models appear applicable in a broader context. The model does not overfit with different coating fractions and complex morphologies. The one-to-one comparison results agree with the results from Lamb and Gentine (2023), which also showed reasonable predictions of $\hat{Q}_{ext}, \hat{Q}_{sca}, \hat{Q}_{abs}, \hat{g}$ across the entire range of size parameters.

During their lifetime, BC fractal aggregates undergo complex changes in size, composition, and morphology due to atmospheric processing. Figure 5 shows a visualization of how the ML predictions compare to the MSTM reference for different aging scenarios for BC fractal aggregates. It compares the estimated and true values of the optical properties for the random split. The models trained using a random split of training data generally show a good agreement with the ground-truth data over the entire range of $D_m$. Overall, the KRR predictions are very close to the true values throughout the entire range of

$D_m$ for all nine cases in Figure 5. The ANN predictions slightly deviate from the true value for cases with larger $f_{\text{coating}}$. For example, in the case of $f_{\text{coating}} = 90\%$ and $D_f = 1.5$, the ANN underestimates the $\hat{Q}_{abs}$. Lamb and Gentine (2023) showed comparatively more deviation in the predictions for larger pure BC fractal particles than smaller particles. In this study, KRR and ANN predictions were consistently good for pure BC fractal particles (first row in Figure 5). Although we could observe

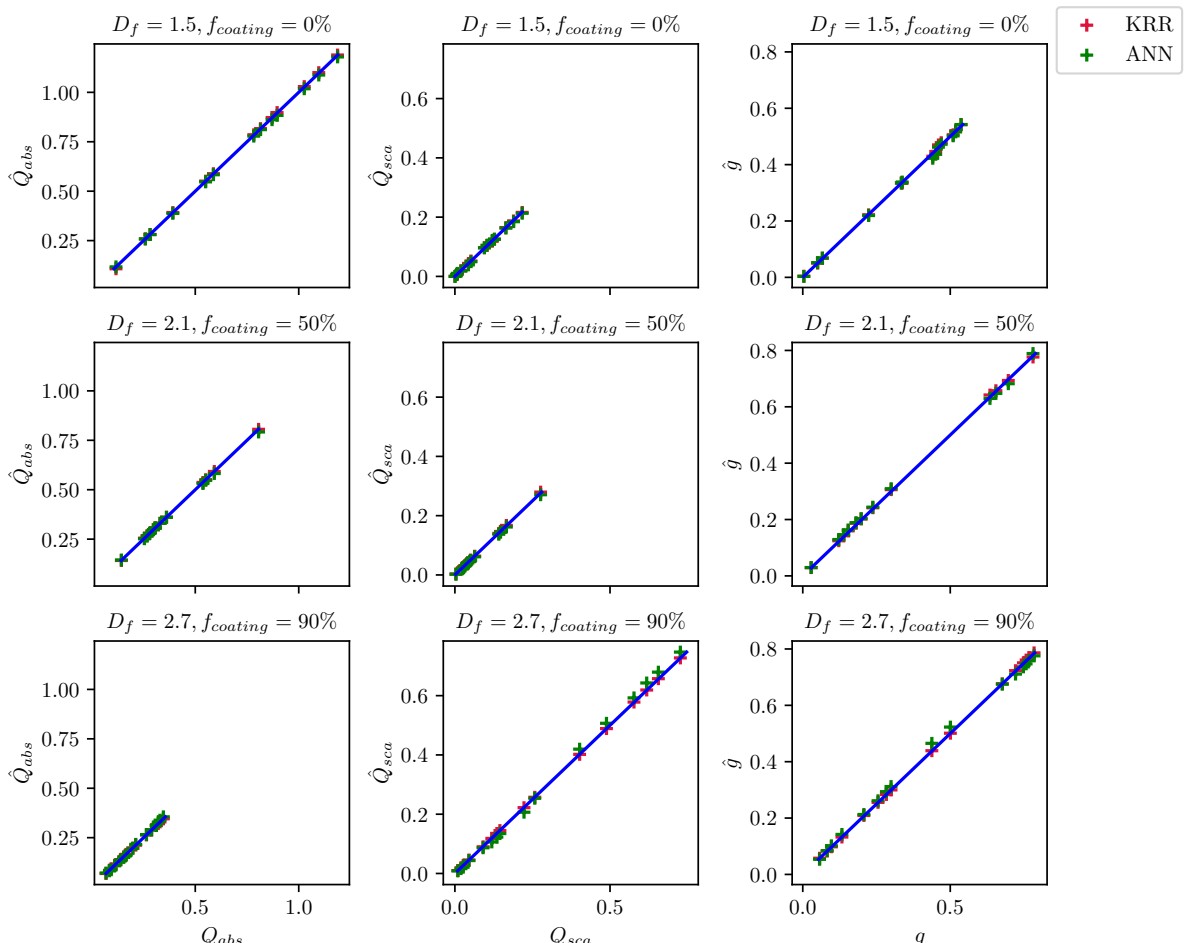

**Figure 4.** Comparison of the predicted optical properties with their true values when the ML models are trained on a random subset of data. The data points for predicted optical properties correspond to KRR and ANN, as shown by the legend on the top right. The blue line in each panel of the figure corresponds to the one-to-one line between the X-axis and Y-axis.

deviations from the true values for large and aged coated particle predictions (last row in Figure 5). Appendix C3 contains

plots similar to Figure 5 for the interpolation and extrapolation split. In general, errors increase with increasing aggregate sizes for the interpolation and extrapolation splits. The ML models we publish are based upon random split experiments, and Fig. 5 shows how well both the ML methods provide accurate estimates of the optical properties of BC fractal aggregates at each aging stage.

    Apart from making accurate predictions, our ML models should also be fast to provide a benefit over time-consuming

simulations. Hence, we recorded the time needed to train on the entire training dataset and the time for making a single prediction in Table 2. As a result, the prediction time of both algorithms is less than one millisecond, which is a drastic

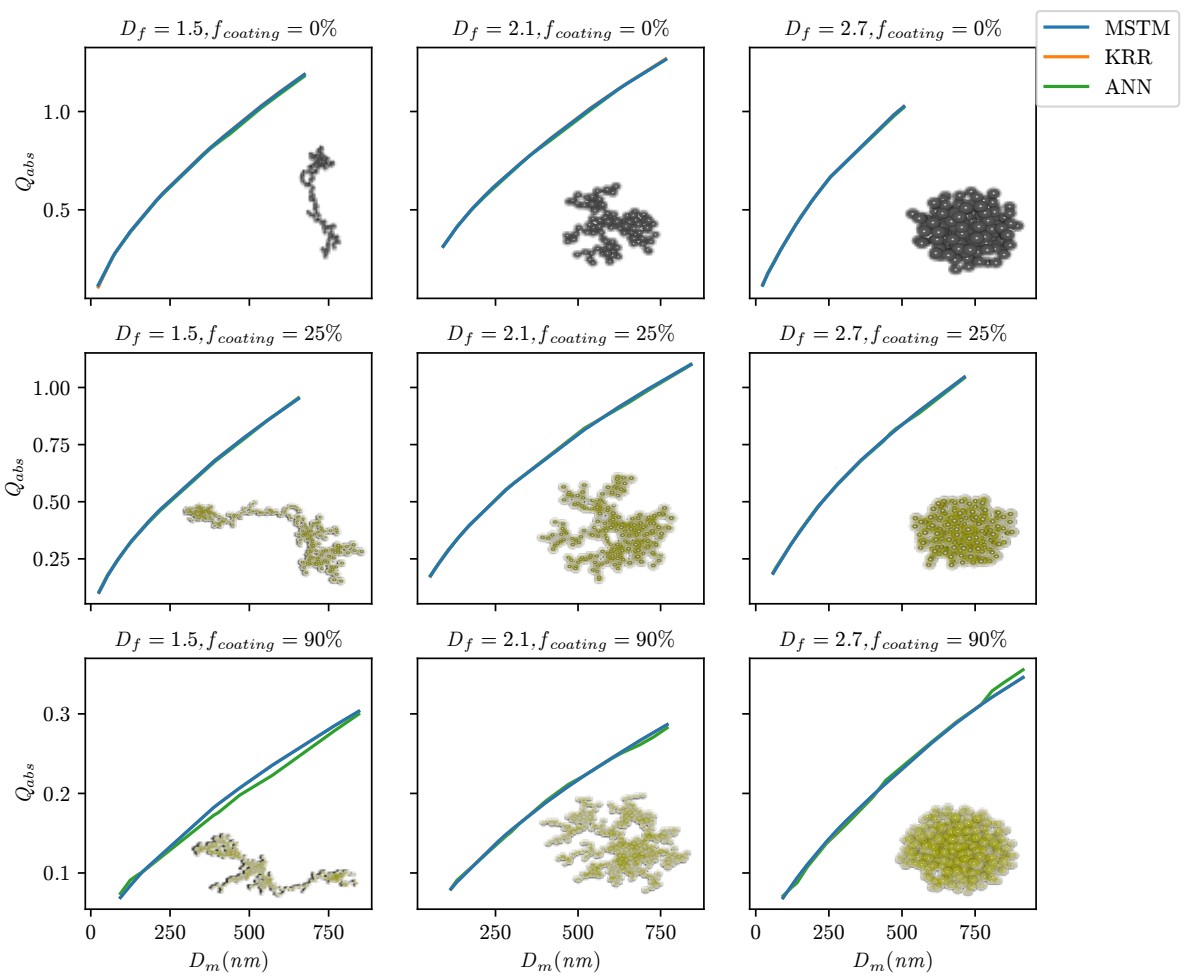

**Figure 5.** Absorption efficiency ($Q_{abs}$) at a wavelength of 660 nm predicted using KRR and ANN for nine representative BC aggregates with a variety of morphologies (represented by $D_f$) and coatings (represented by $f_{coating}$). Both models were trained on a random split of training data.

**Table 2.** Training time for 18526 samples in the dataset and prediction time per sample in seconds. Values were recorded on a machine with Intel(R) Core(TM) i7-9750H CPU, 8 GB RAM, and NVIDIA GeForce GTX 1650 GPU.

| ML model | Training time | Prediction time |
| --- | --- | --- |
| KRR | 33.3s | 0.0006s |
| ANN | 1770s | 0.0005s |

improvement compared to the MSTM method, which can take up to 24 hours, depending on the particle. It should be noted that the prediction time for the ANN does not depend on the input data. Training the models takes comparatively longer, but it is usually done offline. Therefore, it is irrelevant for users using the pre-trained models we provide for their applications (see Section 8).

## 5    Comparison to black carbon laboratory measurements

Incorporating the fractal morphology of BC in global model calculations is essential as the BC radiative forcing can increase up to 61% compared to a more compact and aged particle (Romshoo et al., 2021). In the atmosphere, BC fractal aggregates are primarily found in conjunction with other aerosol types, such as organic carbon. It is, therefore, more relevant to predict the optical properties of BC fractal aggregates with organic coatings for atmospheric applications. To give an example of applying the ML algorithm to real-world atmospheric research, we predicted the optical properties of laboratory-generated soot for experiments described in the Table. 1 of our previous study (Romshoo et al., 2022).

The ML-based predictions were compared to the averages of each experimental case, represented by one data point in Figure 6. The ML results correspond to KRR, the default algorithm used in the prediction script. The details of the laboratory experiments and instrumentations are given in Appendix D. Figure 6A compares the ML algorithm predicted single scattering albedo ($\hat{\omega}_{\mathrm{ML}}$) with the measured $\omega$ from the laboratory experiment. The $\hat{\omega}_{\mathrm{ML}}$ predictions are in good agreement with the measured results for a range of $f_{\mathrm{organics}}$ going up to 55%. The uncertainty of nearly 10% in the measured SSA (Weber et al., 2022) is well represented within the 95% confidence band of the ML-based predictions. On the contrary, Figure 6B demonstrates that if the conventionally used Mie-core-shell theory is used, the predictions are overestimated by a large margin. The ML predictions of *MAC* are also compared to the measured *MAC* and Mie-based predictions, whose results are given in Figure D1 of the Appendix. The predictions $\hat{MAC}_{\mathrm{ML}}$ were found to be less sensitive to the change in $D_{\mathrm{mob}}$. Due to a lack of monodisperse mass measurements, comparing the predictions and measured values is not so straightforward. However, one can see that the discrepancies in the ML-based predictions of *MAC* are comparatively lower than the Mie-derived *MAC* values.

The sensitivity in the predicted *MAC* and *SSA* as a function of change in input parameters such as the $D_{\mathrm{mob}}$, $D_{\mathrm{f}}$, $f_{\mathrm{coating}}$, and $a$ have been extensively discussed by Romshoo et al. (2021, 2023b); Smith and Grainger (2014). The recommendations given by the above studies have been adapted for obtaining the results in Figure 6 and Figure D1, discussed in detail in Appendix D. For future applications, it is recommended to use ambient or laboratory datasets with a resolution of more than 30 minutes to minimize the interference of instrumental uncertainty due to noisy data. Similarly, for ambient or laboratory closure studies, the model output is recommended to be compared with averaged optical observations.

Based on the success of the ML-based approach in predicting the optical properties of coated BC particles, it has great potential for future development to predict the optical properties of mixtures of BC and other aerosols. Because such a study would be exhaustive, we initially tested this approach on BC fractal aggregates and organic coatings to determine its effectiveness. Further research is necessary to develop an ML algorithm with features representing different morphological shapes and other chemical compositions, such as inorganics. In the long run, the goal should be to develop an ML algorithm that can be used

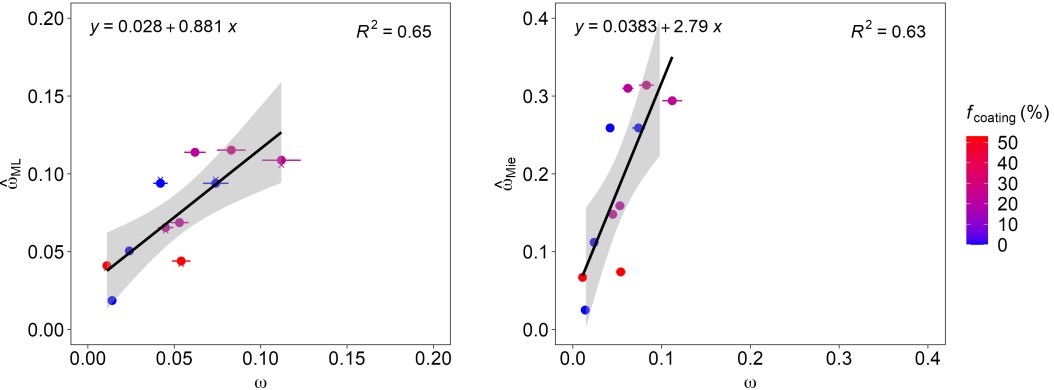

**Figure 6.** Single scattering albedo $\omega$ of coated BC particles at varying $f_{\text{organics}}$, generated in a laboratory study using different miniCAST setpoints (Romshoo et al., 2022). **(A)** compares the $\hat{\omega}_{\text{ML}}$ with the measured $\omega$ from the laboratory experiment. The cross points in the figure show the results from the MSTM-based database used for training the ML algorithm. **(B)** compares the $\hat{\omega}_{\text{Mie}}$ with the measured $\omega$. The ML results correspond to KRR, the default algorithm used in the prediction script. Error bars along the X-axis show the uncertainty in the measured $\omega$. The cross points are the $\omega$ from MSTM simulations. The black linear represents a linear regression equation shown in the upper left corner, with the coefficient of determination ($R^2$) in the upper right corner of each panel. The grey area represents the 95% confidence level interval for predictions.

to integrate all atmospheric aerosols into global climate models. To develop such a universal algorithm for all atmospheric aerosols, we must incorporate the conventional spherically shaped particles into the current prediction algorithm to represent the fraction of aged aerosols. In this study, due to the experimental design of (Romshoo et al., 2022), we could only test the ML-based prediction algorithm for particles with $f_{\text{organics}}$ of less than 65 %. The extension of the current algorithm to include more parameters also demands closure studies using more datasets of laboratory and ambient measurements.

## 6    Limitations and future challenges

The experiments conducted for this study show that our ML methods predict the optical properties of BC fractal aggregates with high accuracy as long as they are trained on sufficient data. However, the interpolation and extrapolation experiments show that the performance of both KRR and ANN significantly deteriorates when entirely removing certain ranges from the

training data. This suggests that our models possess only limited generalization capabilities. Still, it should be noted that we train the models for practical use on the entire physically feasible range of $D_f$ and $f_{\text{coating}}$. Hence, those models will not have to extrapolate for any reasonable inputs.

Our models treat the wavelength $\lambda$ as a continuous variable, meaning they should support computing optical properties at wavelengths that are not part of the training data. The prediction script can predict the optical properties well for the range between 467 and 660, and points close to the upper and lower limit. However, we did not test the models' generalization capabilities about the wavelength since omitting just one wavelength from the training data would reduce the dataset size by one-third. Generating more ground truth data for other wavelengths requires refractive indices of BC and organics for that specific wavelength, which are unavailable in the literature. Even if they were available, it would be time-consuming as MSTM simulations can take a long time to compute. Nevertheless, examining the models' generalization capabilities on other wavelengths in the future would be interesting.

In this study, the ML-based prediction algorithm is developed using training data of $N_{\text{pp}}$ up to 1000, which corresponded to particles with maximum $D_{\text{mob}}$ of 1561 nm depending on the $f_{\text{coating}}$. This range of particle sizes was chosen while designing the database, considering the realistic size of BC-containing particles in the atmosphere. TEM analysis has shown a high probability that the BC-containing particles less than 1500 nm will be fractal ((Adachi et al., 2016; Wang et al., 2017)). The ML algorithm developed in this study, which is based on a close-shell coating model, is suitable for such particles smaller than 1500 nm. However, when aerosol particles grow larger, the mass of BC decreases significantly compared to the mass of coating (Adachi et al., 2016). For such cases of aged BC, using the conventional core-shell-based spherical morphology is appropriate. This is why we limited our training data range for particle size to 1561 nm. However, as demonstrated by Luo et al. (2018a), adding a few points in the training data significantly improves the extrapolation efficiency of machine learning models. Further, some studies show that the optical properties are not sensitive to the change in the primary particle size $a$. Therefore, we fixed the $a_{\text{i}}$ to 15 nm, and changed $a_{\text{o}}$ from 15.1 to 29 depending on the $f_{\text{coating}}$. Similar to the parameters related to a particle size such as $N_{pp}$, $r_o$, and $D_m$, adding a few data points to the $a_{\text{i}} \, or \, a_{\text{o}}$ can help optimize the extrapolation ability of the ML-based prediction algorithm. Although future studies can extend the model's extrapolation ability, the particle size range of the current prediction algorithm covers the physically feasible cases for BC fractal aggregates.

Both KRR and the ANN provide only a single-point prediction for each input. In particular, their estimate does not quantify any uncertainty in the prediction. Bayesian ML methods such as Gaussian Process Regression (Rasmussen and Williams, 2005) can provide information about the uncertainty of a prediction via credible intervals as they return an entire probability distribution instead of a single point estimate. Thus, it would be interesting to examine Bayesian ML for the prediction of BC fractal aggregates' optical properties. This method could be further developed for reporting the predictions for an ensemble of BC-containing aerosols with various physicochemical properties. However, applying them directly to our problem is not trivial since the assumptions made by their statistical model (e.g., target variables follow a multivariate Gaussian distribution) are often violated in practice. Therefore, we leave the application of Bayesian ML to the BC aerosol problem to future work.

Atmospheric BC can exhibit a wide range of morphologies showing diversity at different locations (Sedlacek III et al., 2022). It was observed that aged transported soot can retain its fractal morphology 500 to 1000 km downwind of emission sources (Sun

et al., 2020). The current state of the art for representing atmospheric soot particles focuses on spherical morphology (Aquila et al., 2011; Stier et al., 2005; Bauer et al., 2008). The model provided in this study was designed to simulate the optical properties for the entire BC lifecycle, capturing the transition between fresh fractal and aged spherical particles. Furthermore, the calibration of light absorption measurement devices is mostly done with fresh soot. We can link to atmospheric relevant absorption by simulating mass absorption cross-sections and light absorption enhancement factors. The coating model used in this study is called the "closed-cell model," the results showed good comparability with the realistic coating model (Kahnert, 2017). A more sophisticated coating model would be a good choice, but it requires more complex scattering models, such as Discrete Dipole Approximation (DDA), which is computationally expensive. With the DDA method, generating elaborate datasets for training ML algorithms is not feasible. We provide a method that predicts the optical properties of a wide range of ambient soot particles with high accuracy. Therefore, the results of this study are valuable for the simulation of realistic scenarios, despite the model limitations. There is scope for future studies to extend such an ML-based approach using other morphological models of BC and coating positions.

# 7 Conclusions

The present study demonstrated that the predictions of BC optical properties can be improved by incorporating their realistic morphologies. Unlike the computationally intensive simulations of complex scattering models, the ML-based approach accurately predicts optical properties in fractions of a second. In conjunction with a laboratory dataset, it was shown that optical properties like single scattering albedo $\omega$ and mass absorption cross-section (*MAC*) can be predicted with greater accuracy than with a Mie-based approach. Using an extensive database for the physicochemical and optical properties of BC fractal aggregates, we trained two ML models—KRR and ANN—that can be used to predict the optical properties of coated BC aggregates at all aging stages. In particular, we could accurately predict the optical properties in the visible spectrum for BC fractal aggregates of any desired size, shape, and fraction of organic coating. Thus, this work illustrates the use of this realistic approach in real-world atmospheric research applications.

We summarize the key conclusions of the study as:

- *Active investigation area*: BC is a highly relevant and active field of research, as it affects the climate system and human health. Global climate models require information about the optical properties of BC to simulate their radiative forcing. The BC research will benefit from using this ML algorithm to generate the optical properties of BC based on more realistic fractal aggregates.

- *Broader application*: The ML algorithm can predict the optical properties absorption efficiency, scattering efficiency, and asymmetry parameter for a wide range of BC fractal aggregates with physiochemical properties specified by particle size, morphology, and coating fraction. Previous studies did not consider the critical parameter of coating fraction in their ML models. Therefore, even though we discuss the results in terms of *Number of primary particles* ($N_{pp}$), the user is additionally able to specify the particle size in terms of *Volume equivalent diameter* ($R_v$) or *Mobility diameter* ($D_m$)

depending on numerical or in-situ based nature of the study. We tested the use of the ML algorithm for predicting the scattering properties of laboratory-generated soot particles and found that it was well in agreement with the measured values.

- *User-friendly*: We published a simple Python script that allows users to predict optical properties for BC fractal aggregates using our pre-trained models at GitHub[5]. The user must specify the physicochemical properties of a BC fractal aggregate as a .csv file, from which the prediction script generates the corresponding optical properties using either KRR or ANN.

- *Low computational and energy costs*: Our ML models have a low computational cost, taking fractions of a second to provide the predictions on a run-of-the-mill desktop PC. The same optical properties could take more than 24 hours to be generated when using a T-matrix optical model. Using such ML algorithms will thus reduce the energy expenditures associated with running optical models on supercomputers.

- *Citability and reproducibility*: The dataset used for developing the ML algorithm is available for download at Zenodo (Romshoo et al., 2023b). Furthermore, the baseline experiments can be reproduced with the code that is openly available on GitHub[6].

In summary, we demonstrated the feasibility of incorporating the realistic morphology of BC to improve their predictions of optical properties using a first-of-its-kind machine-learning approach. This ML-based approach constitutes a significant step forward in BC aerosol research in two ways: first, it is the first attempt to provide optical properties of coated BC fractal aggregates at different stages of atmospheric aging using realistic representations. Second, this approach significantly reduces the heavy computational costs of using previous complex scattering models. Previous studies of BC avoid using complex scattering theories because of the high computational costs and prefer the more simplistic Mie theory. This research will be further developed in the future with the final goal of accurately predicting the optical properties of any mixture of atmospheric aerosols. We will investigate if the spherical core-shell model can be combined together with the fractal aggregate-based ML model to distribute the weightage of light absorption predictions for an ensemble of atmospheric BC aerosols with variable aging stages.

# 8  Code and data availability

A Python script that predicts the optical properties of BC fractal aggregates using the trained ML-based models is available in a GitHub repository https://github.com/jaikrishnap/Machine-learning-for-prediction-of-BCFAs (Romshoo et al., 2023d). To run the prediction script, the physio-chemical properties need to be provided as a .csv file that contains the fractal dimension $D_f$, the fraction of coating $f_{\text{coating}}$, and the wavelength ($\lambda$) at which the optical properties should be calculated. Depending on the relevance, users may specify the particle size by giving the values of one among the number of primary particles ($N_{pp}$), the

---

[5]https://github.com/jaikrishnap/Machine-learning-for-prediction-of-BCFAs
[6]https://github.com/jaikrishnap/Optical-properties-of-black-carbon-aggregates

mobility diameter ($D_m$), or the outer volume equivalent radii ($r_o$). If the input parameters are obtained from instrumental measurements, taking hourly or half-hour averages is recommended to cancel the effect of noisy input parameters. The prediction script will generate a .csv file with the corresponding optical properties for the provided physio-chemical properties. Please check the README file inside the repository for more detailed information on using the script.

485  The dataset of simulated physio-chemical and optical properties that we describe in Section 2 is available at https://doi.org/10.5281/zenodo.7523058 (Romshoo et al., 2023b). In case they want to reproduce any of the results in this work, readers may find the entire source code that we used to perform the ML-based experiments and generate figures included in this work on https://github.com/jaikrishnap/Optical-properties-of-black-carbon-aggregates (Romshoo et al., 2023c).

## Appendix A: Details about the physiochemical and optical properties of BC fractal aggregates

**A1  Formulae**

The volume equivalent radius ($r$) is defined as the radius of a sphere having the same volume as the BC fractal aggregate, given as:

$$r = a \sqrt[3]{N_{pp}} \tag{A1}$$

where $N_{pp}$ is the number of primary particles, and $a$ is the radius of a single primary particle. The outer volume equivalent
radius ($r_o$) was calculated for the whole BC aggregate and the coating using $a_o$. The inner volume equivalent radius ($r_i$) was calculated using $a_i$ for the BC aggregate without the coating, i.e., pure BC.

The mobility diameter of a sphere ($D_m$) was defined by Sorensen (2001) as:

$$D_m = 2a_o \left(10^{-2x+0.92}\right) N_{pp}^x, \tag{A2}$$

where $N_{pp}$ is the number of primary particles, and $a_o$ is the radius of a primary particle with coating, $x$ is the mobility mass
scaling exponent given by $x = 0.51 Kn^{0.043}, 0.46 < x < 0.56$. $Kn$ is the Knudsen number, which is the ratio of the molecular free path to the agglomerate mobility radius. The error estimated in the mobility mass scaling exponent ($x$) is $\pm 0.02$.

The relationship between the outer radius of the primary particle ($a_o$), the inner radius of the primary particle ($a_i$), and the fraction of organics ($f_{organics}$) is given as:

$$a_o{}^3 = (1 - f_{organics}) a_i{}^3 \tag{A3}$$

The geometric cross-section ($C_{\text{geo}}$) is the area of the cross-section of the volume-equivalent sphere, given as:

$$C_{\text{geo}} = \pi r_o{}^2 \tag{A4}$$

The optical cross-sections ($C_{\text{ext/abs/sca}}$) are defined as the product of efficiency ($Q_{\text{ext/abs/sca}}$) and geometric cross-section ($C_{\text{geo}}$) as:

$$C_{\text{ext/abs/sca}} = Q_{\text{ext/abs/sca}} C_{\text{geo}} \tag{A5}$$

The asymmetry parameter (or asymmetry factor) $g$ is defined as the average cosine of the scattering angle theta $\theta$:

$$g = \langle cos\theta \rangle \tag{A6}$$

The single-scattering albedo ($\omega$) is derived from the ratio of the scattering efficiency ($Q_{\text{sca}}$) to the extinction efficiency ($Q_{\text{ext}}$) as:

$$\omega = \frac{Q_{\text{sca}}}{Q_{\text{ext}}} \tag{A7}$$

The total mass absorption cross-section ($\text{MAC}_{\text{Total}}$), BC mass absorption cross-section ($\text{MAC}_{\text{BC}}$), and coating mass absorption cross-section ($\text{MAC}_{\text{Coating}}$) were calculated from the ratio of ($C_{\text{abs}}$) with total mass ($m_{\text{Total}}$), BC mass ($m_{\text{BC}}$), and coating mass ($m_{\text{Coating}}$), respectively, as:

$$\text{MAC}_{\text{total/BC/coating}} = \frac{C_{\text{abs}}}{m_{\text{total/BC/coating}}} \tag{A8}$$

## A2  Range of features and constants

**Table A1.** Features from the database of physicochemical and optical properties of black carbon fractal aggregates. For independent features, the list of values are provided. The features for which the range has provided correspond to dependent features.

| Parameter | Values/Range |
|---|---:|
| Wavelength ($\lambda$) | 467, 530, 660 |
| Fractal dimension ($D_f$) | 1.5, 1.7, 1.9, 2.1, 2.3, 2.5, 2.7, 2.9 |
| Fraction of coating ($f_{\text{coating}}$) | 0, 1, 5, 10, 15, 20, 25, 30, 40, 50, 60, 70, 80, 90 |
| Primary particle size ($a_o$) | 15.1 - 29 |
| Number of primary particles ($N_{pp}$) | 1, 2, 3, 4, 5, 6, 7, 8, 9, 10, 12, 14, 16, 18, 20, 23, 26, 29, 31, 34, 36, 39, 42, 45, 50, 55, 60, 65, 70, 75, 85, 95, 105, 115, 125, 140, 155, 170, 185, 200, 225, 250, 275, 300, 350, 400, 450, 500, 550, 600, 650, 700, 800, 900, 1000 |
| Outer volume equivalent radius ($r_o$) | 12 - 290 |
| Inner volume equivalent radius ($r_i$) | 12 - 150 |
| Mobility diameter ($D_m$) | 17 - 1561 |
| Extinction cross-section ($C_{\text{ext}}$) | 0.043 - 3.02 |
| Absorption cross-section ($C_{\text{abs}}$) | 0.041 - 1.75 |
| Scattering cross-section ($C_{\text{sca}}$) | 0.00038 - 1.82 |
| Asymmetry parameter ($g$) | 0.00036 - 0.91 |
| Single scattering albedo (SSA) | 0.00030 - 0.776 |
| Mass absorption cross-section (MAC) | 3.89 - 24.5 |

**Table A2.** Refractive indices (both real and imaginary parts) of BC and organics at various wavelengths in the visible range (Kim et al., 2015).

| Parameter | Wavelength (nm) | | |
|---|---|---|---|
| | 467 | 530 | 660 |
| $n_{\text{BC}}$ | 1.92 | 1.96 | 2.00 |
| $k_{\text{BC}}$ | 0.67 | 0.65 | 0.63 |
| $n_{\text{coating}}$ | 1.59 | 1.47 | 1.47 |
| $k_{\text{coating}}$ | 0.11 | 0.04 | 0.00 |

# Appendix B: Details about the machine learning methods

**Table B1.** Previous machine-learning studies

| Feature | Lamb and Gentine (2023) | Luo et al. (2018) | This study |
|---|---|---|---|
| Machine-learning method | Graph Neural Network (GNN) | Support Vector Model (SVM) | Kernel Ridge Regression (KRR), Artificial Neural Network (ANN) |
| Particle generation | Cluster-cluster algorithm | Tunable diffusion limited algorithm | Tunable diffusion limited algorithm |
| Wavelength | 450, 650 nm | 500 – 3000 nm | 467, 530, 660 nm |
| Outer primary particle size ($a_o$) | 7 – 104 nm | 40 nm | 30 - 60 nm |
| Number of primary particles ($N_{pp}$) | 8 – 960 | 8 – 3000 | 1 - 1000 |
| Fractal dimension ($D_f$) | 1.8 - 2.3 | 1.8 - 2.2 | 1.5 - 2.9 |
| Fraction of organics ($f_{organics}$) | 0 % | 0 % | 0 - 90 % |
| Predictors | $Q_{ext}, Q_{sca}, Q_{abs}, g$ | $Q_{ext}, Q_{sca}, Q_{abs}, g$ | $Q_{ext}, Q_{sca}, Q_{abs}, g, MAC_{BC}, SSA$ |
| Performance metrics | Mean absolute percentage error (MAPE) | Relative error | Mean absolute error (MAE) |
| Comparison to measurements | No | No | Yes |

**Table B2.** Hyperparameter values for the kernel ridge regression (KRR) experiments along with optimal value for each parameter.

| Parameter | Values | Optimal value |
|---|---|---|
| RBF kernel bandwidth ($\gamma$) | $0.0001, 0.001, 0.01, 0.05, 0.1,$ $0.5, 0.75, 1$ | $0.5$ |
| Regularization coefficient ($\lambda$) | $0.0001, 0.001, 0.01, 0.05, 0.5,$ $0.75, 1$ | $0.0001$ |

**Table B3.** Hyperparameter values for the multi-layer perceptron (MLP) experiments along with optimal value for each parameter.

| Parameter | Values | Optimal value |
|---|---|---|
| Number of layers ($L$) | $3, 4, \ldots, 12$ | $6$ |
| Number of neurons ($D^{(l)}$) | $1, 8, 16, 32, 64, 128, 256, 512, 1024$ | $256$ |
| Activation function ($\sigma^{(l)}$) | id, ReLU, Sigmoid[7], tanh, ELU (Clevert et al., 2016), Leaky ReLU (Maas et al., 2013) | ReLU |
| Optimizer | SGD, Adam (Kingma and Ba, 2015), RMSProp[8] | Adam |
| Learning rate | $0.001, 0.005, 0.075, 0.01, 0.05, 0.075, 0.1$ | $0.001$ |
| Loss function ($\mathcal{L}$) | MSE, MAE, Huber, LogCosh[9] | MSE |

**Table B4.** Training range and test range of the features during the interpolation split.

| Feature | Range | Test range | Training range |
|---|---|---|---|
| $D_f$ | $[1.5, 2.9]$ | $[2.1, 2.5]$ | $[1.5, 2.1) \cup (2.5, 2.9]$ |
| $f_{\text{coating}}$ | $[0, 90]$ | $[35, 50]$ | $[0, 35) \cup (50, 90]$ |

**Table B5.** Training range and test range of the features during the extrapolation split.

| Feature | Range | Test range | Training range |
|---|---|---|---|
| $D_f$ | $[1.5, 2.9]$ | $[2.5, 2.9]$ | $[1.5, 2.5)$ |
| $D_f$ | $[1.5, 2.9]$ | $[1.5, 1.9]$ | $(1.9, 2.9]$ |
| $f_{\text{coating}}$ | $[0, 90]$ | $[75, 90]$ | $[0, 75)$ |
| $f_{\text{coating}}$ | $[0, 90]$ | $[0, 15]$ | $(15, 90]$ |

**Table B6.** Maximum errors of different splits for their test sets.

| Optical property | Random split | | Interpolation split | | Extrapolation split | | Feature range |
|---|---|---|---|---|---|---|---|
| | KRR | ANN | KRR | ANN | KRR | ANN | |
| $Q_{abs}$ | 0.17 | 0.34 | 0.38 | 0.34 | 0.23 | 0.21 | $0-2$ |
| $Q_{sca}$ | 0.14 | 0.17 | 0.32 | 0.44 | 0.55 | 1.42 | $0-2$ |
| $g$ | 0.14 | 0.22 | 0.46 | 0.44 | 0.42 | 0.32 | $0-1$ |

## Appendix C: Additional figures

### C1 Error Boxplots

Fig. C1 shows the residuals for the machine-learning methods for the three splits related to the feature $f_{\text{coating}}$: random, extrapolation (training data $f_{\text{coating}} = [0, 75)$), and interpolation (training data: $f_{\text{coating}} = [0, 35) \cup (50, 90]$. When the training and testing data is randomly split, we see that residual errors are concentrated near zero for all intervals of $f_{\text{coating}}$ similar to Fig. 3. The errors from the KRR and ANN are comparable in the random split. For the case of interpolation split, the errors from both ANN and KRR models are comparatively higher for all the three optical properties, i.e., $Q_{abs}$, $Q_{abs}$, and $g$. It was noted in the errors from the interpolation split that KRR performs better for predicting the $Q_{abs}$, whereas the ANN performs better for $g$ predictions. The errors in the $Q_{abs}$, $Q_{abs}$, and $g$ from the extrapolation split were the highest. The error is largest for the predictions when $f_{\text{coating}} = 90$, which is the case farthest away from the training data during an extrapolation split. The relative performance of the ANN and KRR are comparable to those observed in the interpolation split.

### C2 Point-wise comparison of predicted and true values

Fig. C3 and Fig. C4 compare the machine-learning predictions to their true values for the cases where the data was excluded while training the ML model. In the Fig. C3, ML predictions were made after removing the intermediate values of the $D_f$ feature (i.e., 2.1, 2.3, 2.5) from the training data. It was observed that the predictions $\hat{Q}_{abs}$ fitted well with the true values, especially for the KRR method. However, the predictions $\hat{Q}_{sca}$ fluctuate from the true value $Q_{sca}$ as the approach maximum values above 1. For predictions $\hat{g}$, the ML methods ANN and KRR perform slightly differently. In the case of extrapolation split, as shown in Fig. C4, the predictions deviated from their true values for $D_f = 2.7, 2.9$ since the ML models did not see the data. However, we can see that for $D_f = 2.5$ (first row), all the predictions are in better agreement with their true values since it was present in the training data. The predictions $\hat{Q}_{abs}$ and $\hat{Q}_{sca}$ showed reasonable agreement in the case of $D_f = 2.7$. The predictions $\hat{Q}_{sca}$ for the unseen $D_f$ features were observed to be smaller than their true values. The predictions $\hat{Q}_{abs}$, $\hat{Q}_{sca}, \hat{g}$ are most inconsistent with their true values when the $D_f = 2.9$ which is the case farthest away from the training data. Therefore, it is demonstrated that there is comparatively higher uncertainty for predicting optical properties for features out of the range of the training data. Further, the performance of the KRR and ANN varied for different optical properties in such cases of interpolation and extrapolation split. The interpolation split performed better for predicting the optical properties out

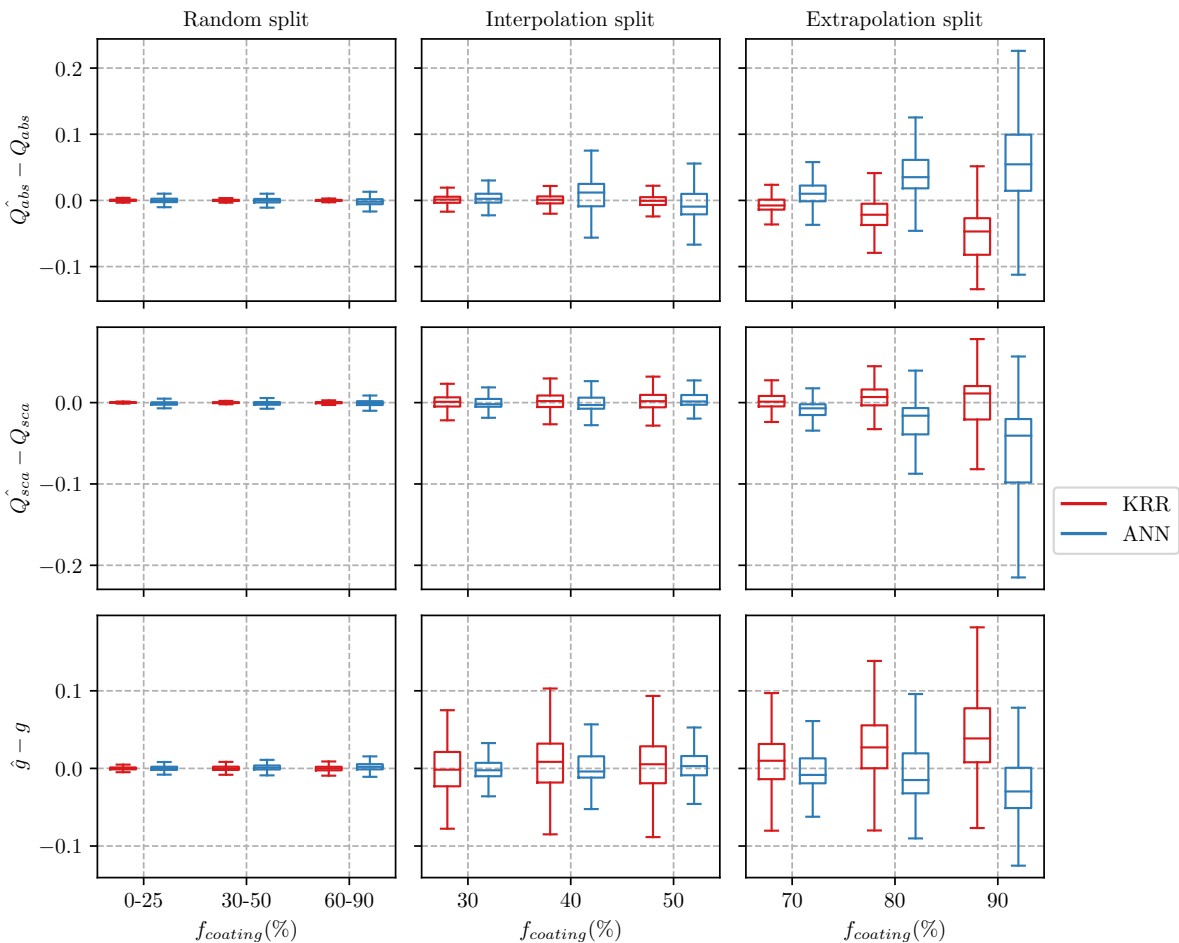

**Figure C1.** Error between the predicted and true values for three optical properties. The residuals are shown when models are trained on data with different ranges of fractions of coating ($f_{\text{coating}}$). The residuals for both KRR and NN predictions are presented in each panel.The lower hinge and the upper hinge of the boxplot represent the 25 % and 75 % quantile of the observations, respectively. Note that the outliers significantly reduced the visualization of the boxplots and, therefore, were omitted from the figures. However, please all the outliers are considered in the training data and error evaluation.

of the range of the training data. Therefore, adding more data in the training set for boundary values to let it interpolate would result in better predictions.

## C3 Line plots showing performance as aggregate size changes

Fig. C5 compares the machine-learning predictions to their true values for interpolation split. The predictions for the case 550 $D_f = 2.3$ (middle row) showed the highest deviations from the true values since it is the farthest point in the training data for the interpolation split. From the $\hat{Q}_{abs}$ results, the KRR predictions were reasonable for the entire size range. The predictions

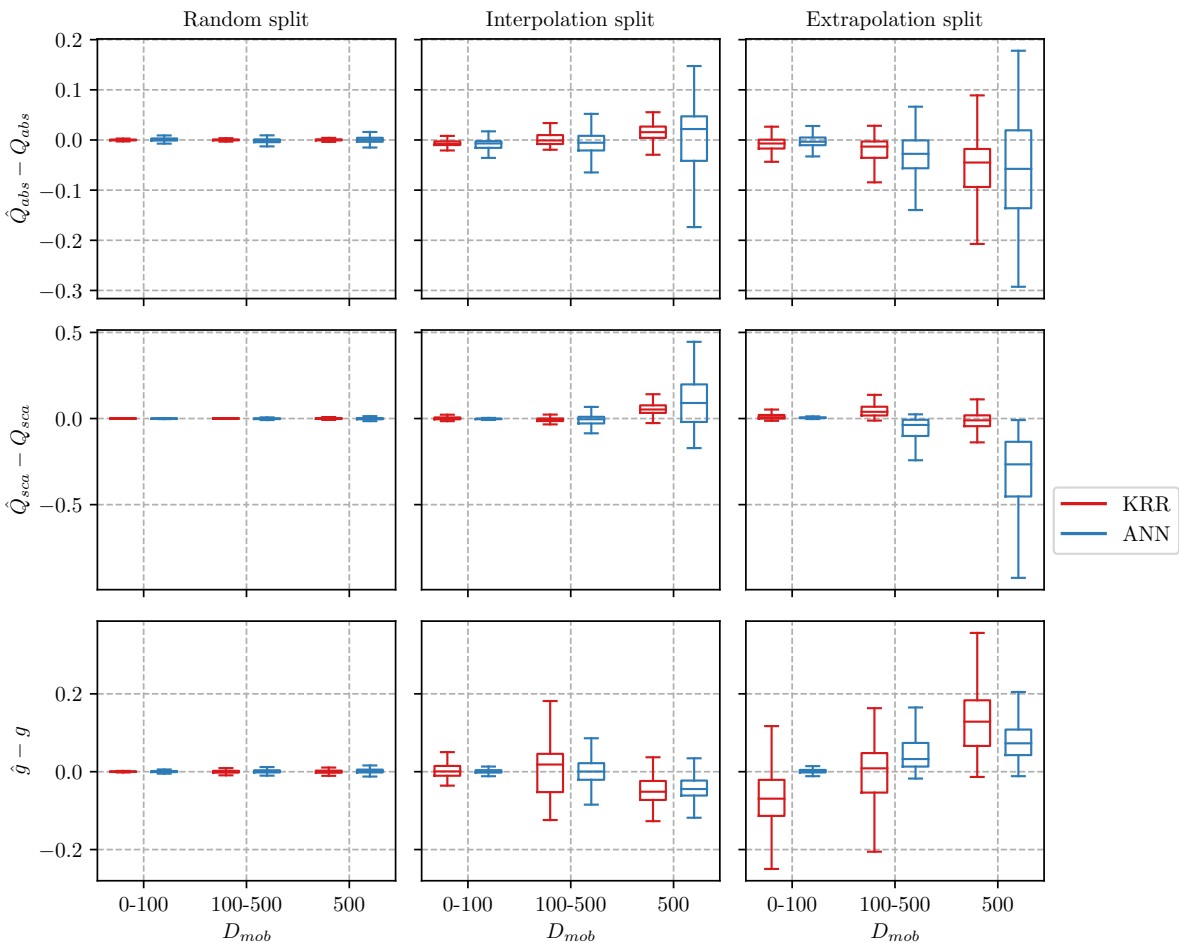

**Figure C2.** Error between the predicted value ($\hat{Q}_{\text{abs}}, \hat{Q}_{\text{sca}}, \hat{g}$) and the true value for three optical properties for various cases of mobility diameter ($D_{\text{mob}}$). The lower hinge and the upper hinge of the boxplot represent the 25 % and 75 % quantile of the observations, respectively. Note that the outliers significantly reduced the visualization of the boxplots and, therefore, were omitted from the figures. However, please all the outliers are considered in the training data and error evaluation.

for $\hat{Q}_{sca}$ were also reasonable for KRR. However, after the particle size increased to larger than 500 nm, the prediction of $\hat{Q}_{sca}$ using KRR was underpredicted. The prediction of $\hat{Q}_{sca}$ using ANN showed a size-dependent behavior, under-predicting the results for certain particle sizes, after which there is an over-prediction. Similar size-dependent behavior was observed in the

predictions $\hat{g}$ from ANN and KRR. The $\hat{g}$ predictions showed deviations from their true values as the particle size increased. In the case of interpolation split, the overfitting or underfitting is generally more pronounced in the larger particle size (> 500 nm). The explanation for this could be the lower resolution of the training data for particle size > 500 nm, which was a limitation of large computation time for larger particles and more coating fraction.

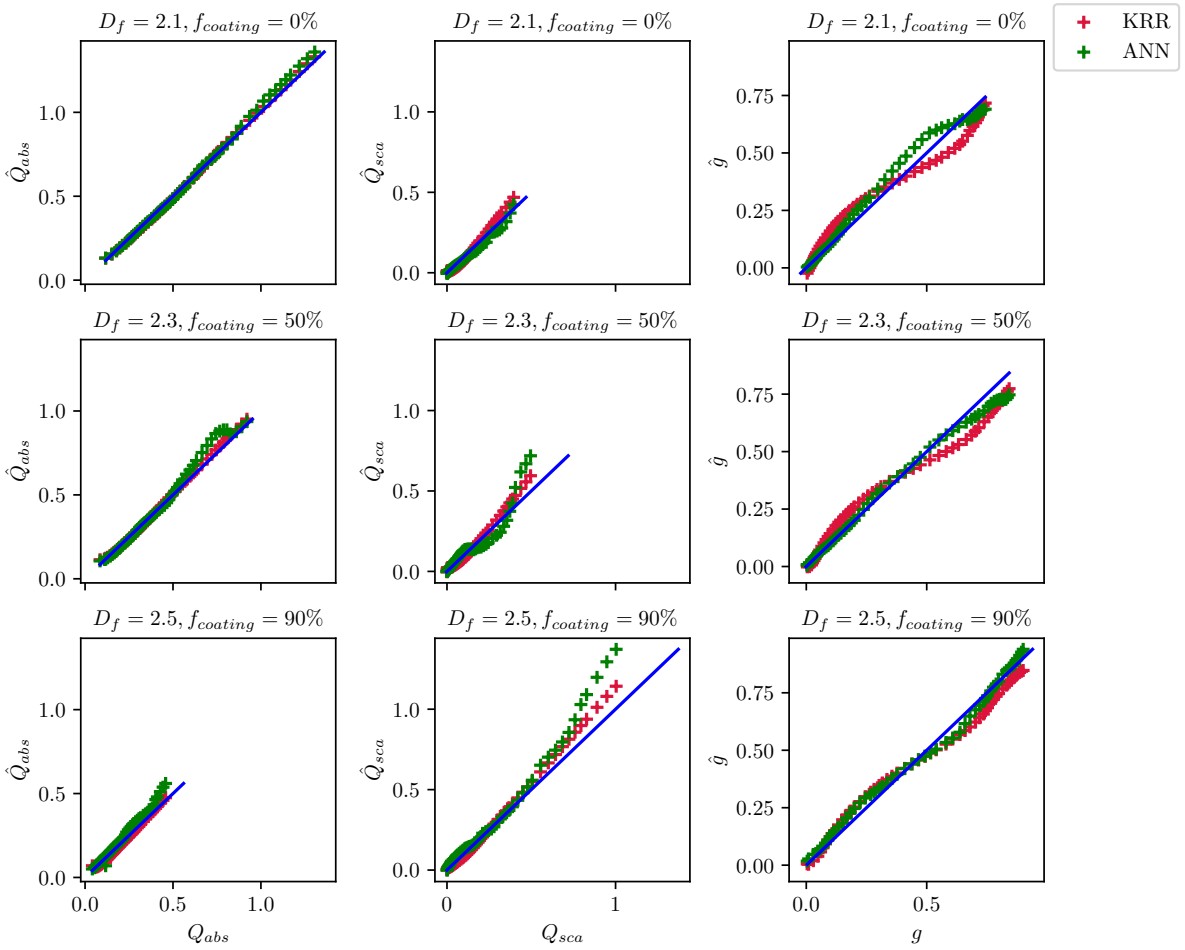

**Figure C3.** Comparison of the predicted optical properties with their true values for the interpolation split when the ML models have trained on data with boundary fractal dimensions: $D_f = 1.5, 1.7, 1.9, 2.7, 2.9$ and tested the model on data with inner fractal dimensions: $D_f = 2.1, 2.3, 2.5$).

Similarly, Fig. C6 show the machine-learning predictions compared to the true values for the extrapolation split. To study
the performance of the KRR and ANN, the results for $D_f = 2.9$ are interesting since they are the farthest from the training
data. The deviations of the $\hat{Q}_{abs}$ are more from the true values in the case of the KRR, which showed better performance in
the interpolation-split. However, the results for $D_f = 2.5$ and $D_f = 2.7$ show reasonable results since they are closer to the
training data set. The predictions $\hat{Q}_{sca}$ were lower than the true values for ANN, especially as the particle size increased. The
prediction $\hat{g}$ was larger than its true value in the case of extrapolation-split. However, the performance of predicting $\hat{g}$ from
KRR showed an interesting size dependence over particle size unique to this split. When particle sizes were smaller, $\hat{g}$ was
higher than the true value, decreased, and returned to higher levels once a certain threshold was reached. In general, for the

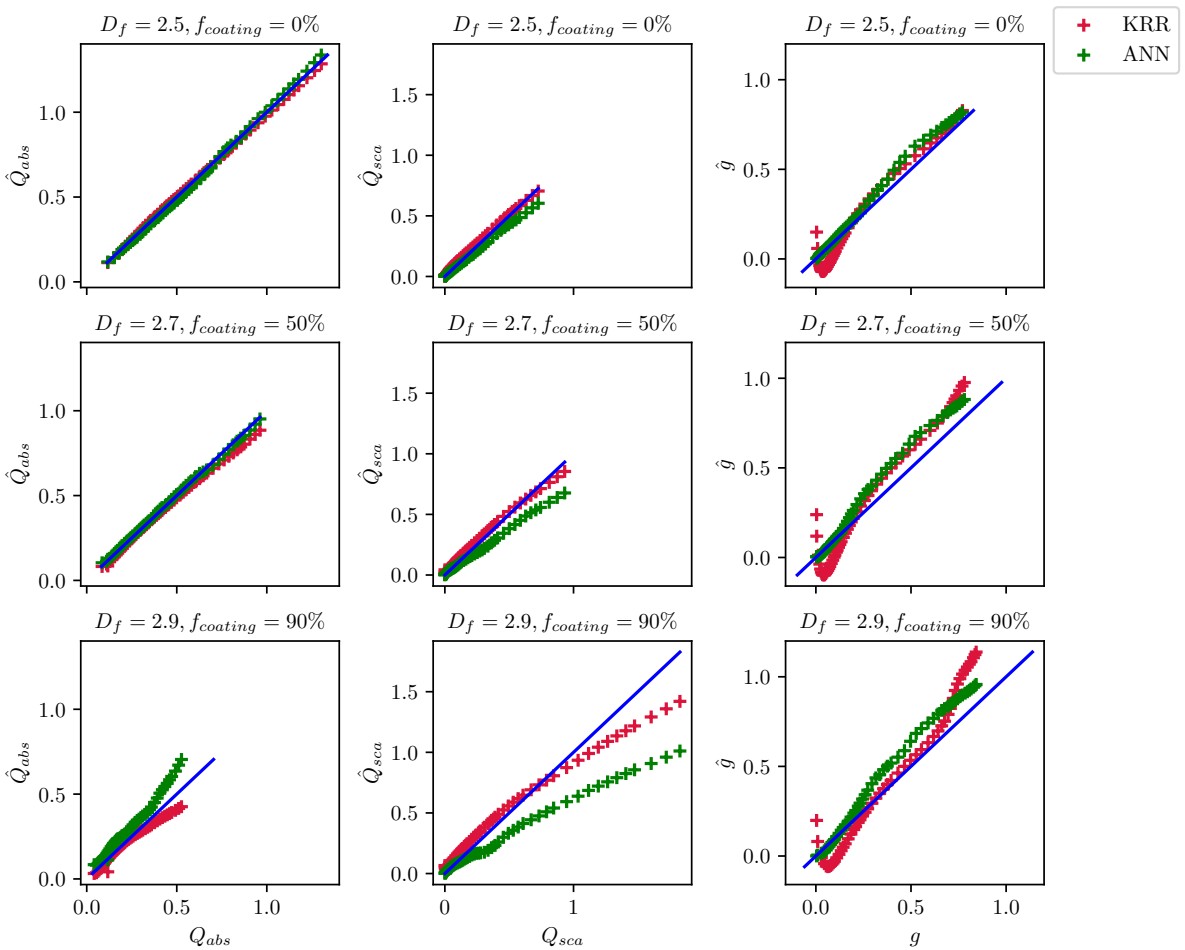

**Figure C4.** Comparison of the predicted optical properties with their true values for extrapolation split when the ML models are trained on data with smaller fractal dimensions: $D_f = 1.5, 1.7, 1.9, 2.1, 2.3$ and tested the model on data with boundary fractal dimensions: $D_f = 2.5, 2.7, 2.9$).

results when the $f_{coating}$ is 90, which is the upper limit of the feature, the results for $\hat{Q}_{abs}$, $\hat{Q}_{sca}$, and $\hat{g}$ showed an expected higher deviation from their true values for both interpolation and extrapolation split.

## Appendix D: Laboratory measurements of black carbon

The data from the laboratory experiments by Romshoo et al. (2022) is compared to the ML-based prediction model in Fig. 6 and Fig. D1. A mobility particle size spectrometer (MPSS, TROPOS – Leibniz Institute for Tropospheric Research – design) measured the particle number size distribution of the black carbon particles. A cavity-attenuated phase shift extinction monitor (CAPS PMex 630, Aerodyne Res. Inc., USA) measured the light extinction coefficient, $\sigma_{ext}$, at a $\lambda$ of 630 nm. The particle

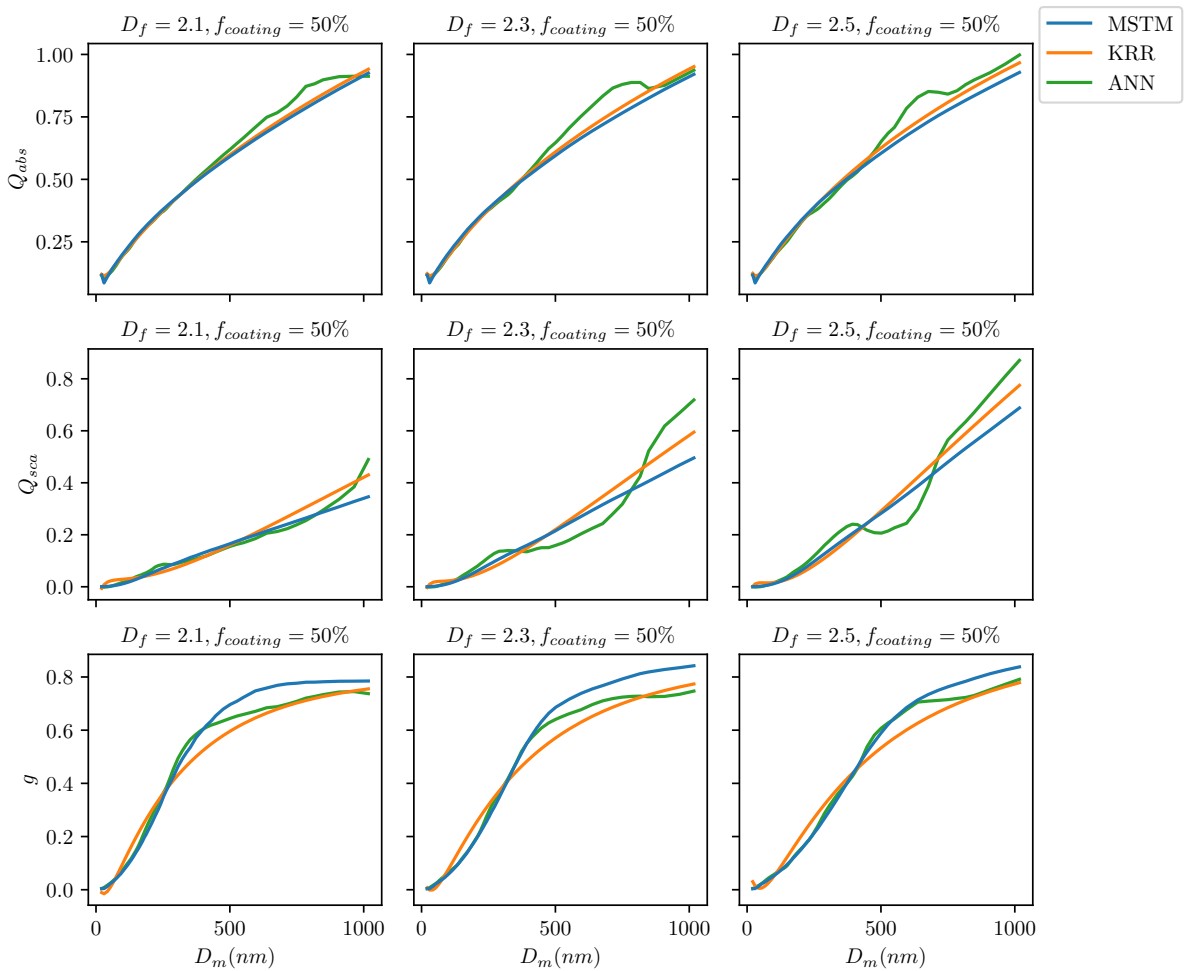

**Figure C5.** Optical properties of BC fractal aggregates predicted using machine learning methods KRR and NN for the interpolation split: when models have trained on data with boundary fractal dimensions $D_f = 1.5, 1.7, 1.9, 2.7, 2.9$ and tested the model if it fits for the intermediate values of fractal dimensions $D_f = 2.1, 2.3, 2.5$).The three columns show the predicted values of absorption efficiency($Q_{abs}$), scattering efficiency($Q_{sca}$), and asymmetry parameter($g$). Each row corresponds to the predictions for the intermediate values of fractal dimensions $D_f = 2.1, 2.3, 2.5$.

light-scattering coefficient $\sigma_{\text{sca}}$ was measured using a nephelometer (Aurora 4000, Ecotech Pvt Ltd, Melbourne, Australia) at $\lambda$
of 635 nm. A multi-angle absorption photometer (MAAP, type 5012, Thermo Scientific, Franklin, MA) measured the particle light-scattering coefficient, $\sigma_{\text{abs}}$, at a $\lambda$ of 637 nm. The aerosol mass concentration for selected experiments was determined using the tapered element oscillating microbalance (TEOM 1405, Thermo Scientific, Franklin, MA). Aerosols were collected on quartz fiber filters and were analyzed by an EC–OC analyzer (Sunset Laboratory Inc., Hillsborough, USA).

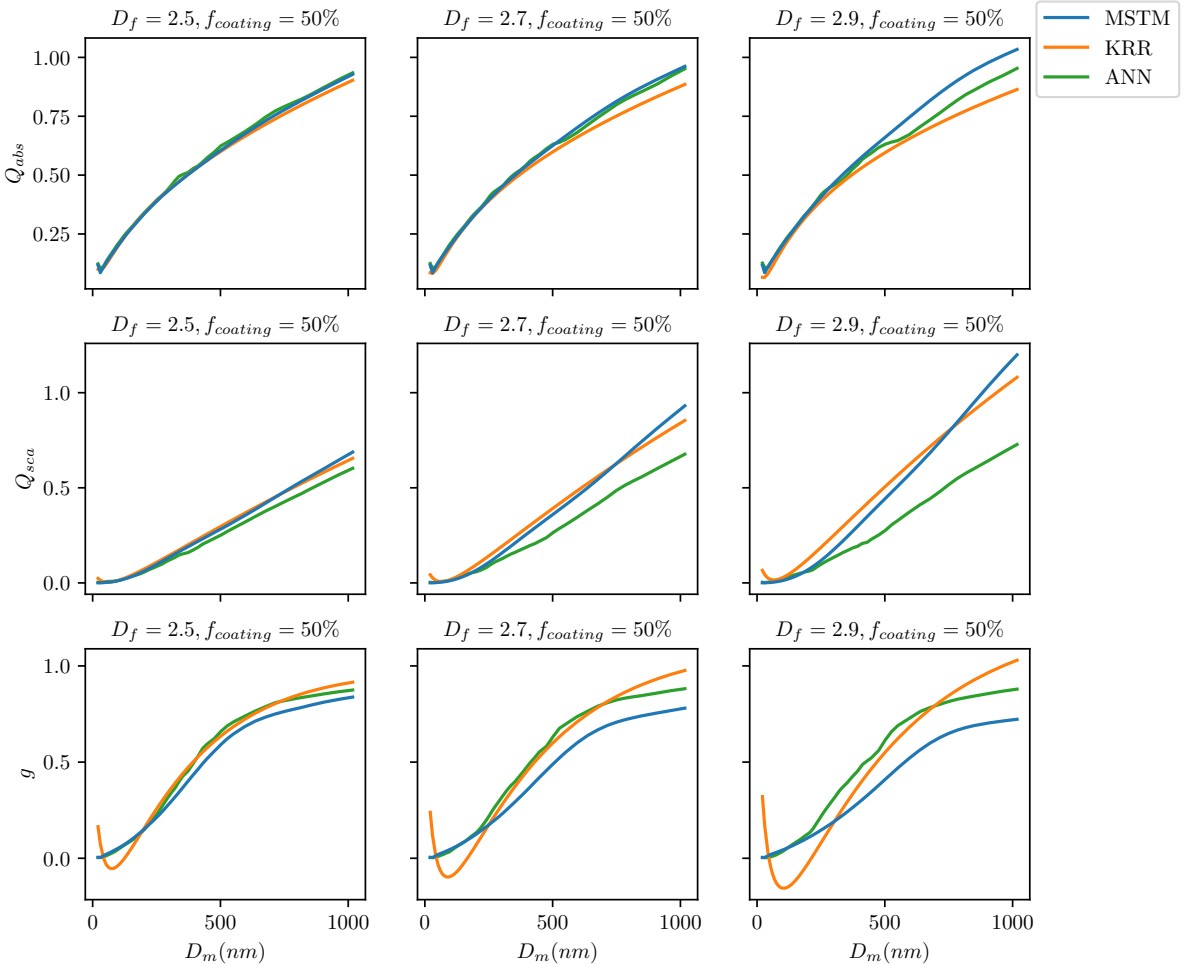

**Figure C6.** Optical properties of BC fractal aggregates predicted using machine learning methods KRR and NN for the extrapolation split: when models are trained on data with smaller fractal dimensions $D_f = 1.5, 1.7, 1.9, 2.1, 2.3$ and tested the model on data with higher fractal dimensions $D_f = 2.5, 2.7, 2.9$).The three columns show the predicted values of absorption efficiency($Q_{abs}$), scattering efficiency($Q_{sca}$), and asymmetry parameter($g$). Each row corresponds to the predictions for the left-out higher fractal dimensions $D_f = 2.5, 2.7, 2.9$.

The input parameters used while running the prediction script are $\lambda$, $D_f$, $f_{\text{coating}}$, and $D_m$. The parameter of $D_m$ was cho-
sen for particle size due to the MPSS measurements available in the experiment. $D_f$ value of 1.7 was taken as it represents
laboratory-generated soot (Wentzel et al., 2003). The default $a_i$ value of 15 nm was used. Numerical studies have also investi-
gated the sensitivity to input parameters like $a$, $D_f$, and $f_{\text{coating}}$ to modeled optical properties (Romshoo et al., 2022; Luo et al.,
2018b; Smith and Grainger, 2014). For example, Romshoo et al. (2022) recommended $D_f$ from 1.7 to 1.9 and $a$ between 10
and 14 nm for laboratory-generated soot. The values of $f_{\text{coating}}$ for each experiment were derived from the EC–OC analysis
results of the quartz fiber filters. The mean of the number size distribution measured by the MPSS was used as the input values
for $D_m$. There were 11 sub-cases of the laboratory experiment for which the mean of $D_m$ and $f_{\text{coating}}$ were taken as input. The
input parameters for the Mie core-shell theory were $\lambda$, $f_{\text{coating}}$ and $D_m$.

The output parameters compared to the observations were $SSA$ and $MAC$. The observational $SSA$ was calculated from the
ratio of $\sigma_{\text{sca}}$ and $\sigma_{\text{ext}}$. The observational $MAC$ was calculated from the $\sigma_{\text{abs}}$ and mass using Eq. (A8). The predicted $SSA$ is
compared to all the 11 experimental cases for which the observational $SSA$ was available (Table 1 in Romshoo et al. (2022)).
The uncertainty in the measured SSA is nearly 10% (Weber et al., 2022). The uncertainties in the SSA are included in the 95%
confidence band of the ML-based predictions. The predicted $MAC$ is compared to the six experimental cases of coated soot
for which the observational $MAC$ was available (last six rows in Table 1 in Romshoo et al. (2022)).

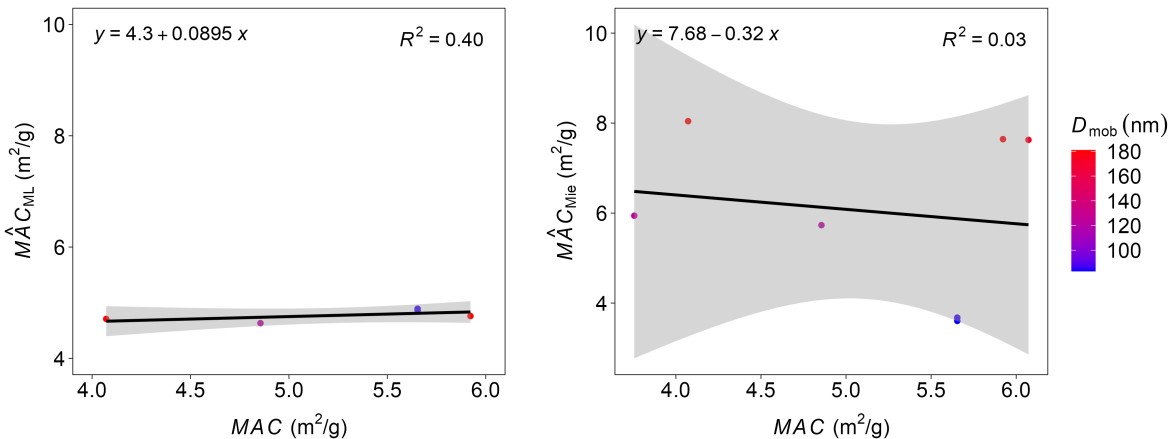

**Figure D1.** Mass absorption cross-section *(MAC)* for coated BC particles generated in a laboratory study at different $D_{mob}$ (Romshoo et al.,
2022). **(A)** compares the $M\hat{A}C_{ML}$ with the measured *MAC* from the laboratory experiment. **(B)** compares the $M\hat{A}C_{Mie}$ with the measured
*MAC*. The number of points used in this figure is less than Fig. 6 as some of the data was excluded due to the uncertainties associated with
the tapered element oscillating microbalance (TEOM) instrument.

*Author contributions.* The study was designed by BR, ThM, JP, ToM, MP, and MK. BR and ThM developed the optical simulations and database. The machine learning experiments were conducted by JP and ToM, with help from BR and ThM. The results were prepared by JP and ToM, with help from BR. The paper was written by BR, JP, and ToM. The paper was reviewed, commented on, and edited by ThM, MK, and MP.

*Competing interests.* The authors declare that none of the authors have any competing interests.

*Acknowledgements.* This research has been supported by the "Metrology for light absorption by atmospheric aerosols" project funded by the European Metrology Programme for Innovation and Research (EMPIR, grant no. 16ENV02 Black Carbon). We would like to thank the members of the European Metrology Programme for Innovation and Research EMPIR 16ENV02 Black Carbon project for their support and feedback. MK acknowledges support by the Carl-Zeiss Foundation, the DFG awards KL 2698/2-1, KL 2698/5-1, KL 2698/6-1, and KL 2698/7-1, and the BMBF awards 03|B0770E and 01|S21010C.

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
