# Peer review of "Improving the predictions of black carbon (BC) optical properties at various aging stages using a machine-learning-based approach"

_EGUsphere, 2023_

## Author Comment (AC1)

Dear Reviewer, we would like to express our gratitude to you for providing valuable comments and suggestions that have significantly improved our manuscript. Below, you will find our point-by-point responses to the individual comments of the two referees, along with a description of the changes incorporated into the manuscript. For clarity, we have used the following color code in the author's response to reviewers:

- Black: the referee's comments.
- Blue: the authors' responses.
- Red: Quotes from the original manuscript.
- *Red & italic: Text modifications in the manuscript related to the referee comments.*

**Major comments:**

**1. Enhanced Contextualization with Previous Studies**: The manuscript should provide a more comprehensive contextualization in relation to prior research. It is mentioned that Luo et al. (2018) and Lamb and Gentine (2021) have previously developed machine learning (ML) algorithms to predict the optical properties of bare black carbon (BC) particles. A more detailed introduction and a comparative analysis with these studies is recommended. Clarify, for example, the range of fractal dimensions and primary sphere sizes used in, as well as the wavelengths for which they trained their models. If possible, employ similar performance metrics as those used in the referenced studies for a direct comparison. Additionally, discuss how the results of your predictions for bare BC compare with previous findings.

**R:** Thank you for the comment. The section about the previous studies was improved by incorporating a better introduction to the previous studies.

[revised manuscript text omitted]

*Figure 5: Overall, the KRR predictions are very close to the true values throughout the entire range of $D_m$ for all nine cases in Figure 5. The ANN predictions slightly deviate from the true value for cases with larger $f_{coating}$. For example, in the case of $f_{coating}$ = 90 % and $D_f$ = 1.5, the ANN underestimates the $Q_{abs}$. Lamb and Gentine (2023), showed comparatively more deviation in the predictions for larger pure BC fractal particles than smaller particles. In this study, KRR and ANN predictions were consistently good for pure BC fractal particles (first row in Figure 5). Although we could observe deviations from the true values for large and aged coated particle predictions (last row in Figure 5).*

**2. Detailed Description and Justification of Model Selection**: Provide a more detailed account of the selected models among those tested, including the rationale for choosing the final two. While Tables B1 and B2 detail different hyperparameters tested, a clear indication of the best and final values or methods selected is necessary. Explain which metric was used to compare the different models and select the best one. Elucidating these choices will significantly enhance the reader's understanding of the research process and the robustness of the final models.

**R:** Thank you for the comment. The following has been added to the manuscript:

*To find a suitable regression model, we conducted experiments with multiple ML-based models for regression, including Support Vector Regression (SVR), Ridge Regression (RR), Kernel Ridge Regression (KRR), and Artificial*

*Neural Network (ANN). Each model was evaluated using Mean Absolute Error (MAE) on the sample dataset. The results showed that Kernel ridge regression and Neural Networks demonstrated better performance, especially in capturing the non-linear relationships within the dataset. Hence, we used KRR and Neural Networks for further analysis.*

We have also updated Tables B2 and B3 (previously B1 and B2, respectively) with another column representing the optimal value chosen for training the model.

**Table B2.** Hyperparameter values for the kernel ridge regression (KRR) experiments along with optimal value for each parameter.

| Parameter | Values | Optimal value |
|---|---|---|
| RBF kernel bandwidth ($\gamma$) | $0.0001, 0.001, 0.01, 0.05, 0.1,$ $0.5, 0.75, 1$ | 0.5 |
| Regularization coefficient ($\lambda$) | $0.0001, 0.001, 0.01, 0.05, 0.5,$ $0.75, 1$ | 0.0001 |

**Table B3.** Hyperparameter values for the multi-layer perceptron (MLP) experiments along with optimal value for each parameter.

| Parameter | Values | Optimal value |
|---|---|---|
| Number of layers ($L$) | $3, 4, \ldots, 12$ | 6 |
| Number of neurons ($D^{(l)}$) | $1, 8, 16, 32, 64, 128, 256, 512, 1024$ | 256 |
| Activation function ($\sigma^{(l)}$) | id, ReLU, Sigmoid[6], tanh, ELU (Clevert et al., 2016), Leaky ReLU (Maas et al., 2013) | ReLU |
| Optimizer | SGD, Adam (Kingma and Ba, 2015), RMSProp[7] | Adam |
| Learning rate | $0.001, 0.005, 0.075, 0.01, 0.05, 0.075, 0.1$ | 0.001 |
| Loss function ($\mathcal{L}$) | MSE, MAE, Huber, LogCosh[8] | MSE |

**3. Model Testing and Regularization Considerations**: Discuss whether regularization techniques, such as dropout or data augmentation, were employed to improve the generalization of your models during testing for both interpolation and extrapolation. As the ultimate goal is to apply these algorithms to ambient data, please clarify if any steps were taken to incorporate noise into the training data to simulate the typical real-world measurement errors.

**R:** Thank you for the comment. The machine learning models KRR and ANN had a default generalization feature used in this study. For the interpolation and extrapolation splits, we carefully designed six experimental scenarios focusing on the parameters of $f_{coating}$ and $D_f$. Tables B4 and B5 provide the details of the experimental design to test the interpolation and extrapolation.

The predictions made by the ML algorithms are quite linear in nature, as can be seen from Figure 5, which was used to reproduce the aging scenarios of BC fractal particles. Therefore, we did not use dropout or data augmentation when designing the prediction algorithm. The ML- algorithm provided in this study is a forward problem that uses noise-free data from MSTM simulations. Our main goal was to provide an accurate and fast ML-based optical model for calculating the optical properties of BC fractal aggregates. However, when using input data from measurements, it is recommended to avoid datasets with a resolution of less than 30 minutes to cancel the noise due to instrumental uncertainty. Similarly, for ambient or laboratory closure studies, the model output is recommended to be compared with averaged optical observations. Incorporating noise in the predictions is highly relevant for inverse problems,

where machine-learning techniques are based on observational input data. The above has been discussed in the revised manuscript as:

*In the case of Kernel ridge regression, regularization is carried out by the regularization constant lambda $\lambda$ (chosen optimal value = 0.0001). In the case of Neural Networks, we did try to use the dropout technique to prevent overfitting. However, dropout regularization did not show notable improvements in the model's generalization.*

*The ML-based predictions were compared to the averages of each experiment which is represented by one data point in Figure 6. The details of the laboratory experiments and instrumentations are given in Appendix D. For future applications, avoiding datasets with a resolution of less than 30 minutes is recommended to avoid interference of instrumental uncertainty due to noisy data. Similarly, for ambient or laboratory closure studies, the model output is recommended to be compared with averaged optical observations.*

**4. Instrumentation and Error Analysis**: Section 5 should be expanded with a brief description of the measurement methods. Although these are described elsewhere, detailing the instruments used, including their associated errors, will introduce the reader to the uncertainties associated with the measurements. How were the input parameters (e.g., the number of primary spheres and fractal dimension) for the ML models and Mie calculations determined? How the measurement errors influence the predicted values?

**R:** Thank you for the comment. The details about the laboratory experiment, input parameters, and measurement errors are given as:

*The data from the laboratory experiments by Romshoo et al. (2022) is compared to the ML-based prediction model in Fig. 6 and Fig. D1. A mobility particle size spectrometer (MPSS, TROPOS – Leibniz Institute for Tropospheric Research – design) measured the particle number size distribution of the black carbon particles. A cavity-attenuated phase shift extinction monitor (CAPS PMex 630, Aerodyne Res. Inc., USA) measured the light extinction coefficient, $s_{ext}$, at a $\lambda$ of 630 nm. The particle light-scattering coefficient $s_{sca}$ was measured using a nephelometer (Aurora 4000, Ecotech Pvt Ltd, Melbourne, Australia) at $\lambda$ of 635 nm. A multi-angle absorption photometer (MAAP, type 5012, Thermo Scientific, Franklin, MA) measured the particle light-scattering coefficient, $s_{abs}$, at a $\lambda$ of 637 nm. The aerosol mass concentration for selected experiments was determined using the tapered element oscillating microbalance (TEOM 1405, Thermo Scientific, Franklin, MA). Aerosols were collected on quartz fiber filters and were analyzed by an EC–OC analyzer (Sunset Laboratory Inc., Hillsborough, USA).*

*The input parameters used while running the prediction script are $\lambda$, $D_f$, $f_{coating}$, and $D_m$. The parameter of $D_m$ was chosen for particle size due to the MPSS measurements available in the experiment. The mean of the number size distribution measured by the MPSS was used as the input values for $D_m$. $D_f$ value of 1.7 was taken as it represents laboratory-generated soot (Wentzel et al., 2003). The default $a_i$ value of 15 nm was used. Numerical studies have also investigated the sensitivity of input parameters like $a$, $D_f$, and $f_{coating}$ to modeled optical properties (Romshoo et al., 2022; Luo et al., 2018b; Smith and Grainger, 2014). For example, Romshoo et al. (2022) recommended $D_f$ from 1.7 to 1.9 and between 10 and 14 nm for laboratory-generated soot. The values of $f_{coating}$ for each experiment were derived from the EC–OC analysis results of the quartz fiber filters. There were 11 sub-cases of the laboratory experiment for which the mean of $D_m$ and $f_{coating}$ were taken as input. The input parameters for the Mie core-shell theory were $\lambda$, $f_{coating}$, and $D_m$. For the closure study, we took the averages for all the observational input parameters to avoid interference from noisy data.*

*The output parameters compared to the observations were SSA and MAC. The observational SSA was calculated from the ratio of $s_{ext}$ and $s_{sca}$. The observational MAC was calculated from the $s_{abs}$ and mass using Eq. (A8). The predicted SSA is compared to all the 11 experimental cases for which the observational SSA was available (Table 1 in Romshoo et al. (2022)).*

*The predicted MAC is compared to the six experimental cases of coated soot for which the observational MAC was available (last six rows in Table 1 in Romshoo et al. (2022)).*

Uncertainty in the SSA measurements is included along the X-axis in the Figure 6 as below:

*The uncertainty in the measured SSA is nearly 10% (Weber et al., 2022). The uncertainties in the SSA are included in the 95% confidence band of the ML-based predictions.*

[Figure]

*Figure 6. Single scattering albedo ω for coated BC particles generated in a laboratory study at different $f_{coating}$ (Romshoo et al., 2022). (A) compares the ωML with the measured ω from the laboratory experiment. (B) compares the ωMie with the measured ω. The ML results correspond to KRR, the default algorithm used in the prediction script. Error bars along the X-axis show the uncertainty in the measured ω. The cross points are the ω from MSTM simulations. The black linear represents a linear regression equation shown in the upper left corner, with the coefficient of determination in the upper right corner of each panel. The grey area represents the 95% confidence level interval for predictions.*

**Minor comments:**

**MC 1:** Abstract: Please provide a detailed and explicit list of the input parameters used in the developed machine learning algorithms.

**R:** Thank you for the comment. An explicit list of the input parameters was added to the Abstract:

*Physiochemical properties of BC, such as total particle size (number of primary particles ($N_{pp}$), outer volume equivalent radius ($r_o$), and mobility diameter ($D_m$), outer primary particle size ($a_o$), fractal dimension ($D_f$), wavelength (l), the fraction of coating ($f_{coating}$) were used as input parameters for the developed ML-algorithm.*

**MC 2:** L5: To substantiate the claim of 'highly accurate,' please provide a specific, quantifiable metric or set of metrics that define and measure the accuracy level being referred to.

**R:** Thank you for the comment. The following sentence explaining the error values was added to the Abstract:

*The mean absolute errors (MAE) for the optical efficiencies ranged between 0.0019 and 0.0039, whereas they ranged between 0.0038 and 0.0044 for the asymmetry parameter.*

**MC 3:** L8-12: In my opinion, the focus of this paper is not demonstrating the superiority of MSTM simulations over Mie theory, as was already shown in Romshoo et al. (2022). Instead, it illustrates how machine learning methods can be employed to predict the optical properties of black carbon particles based on MSTM simulations.

**R:** Thank you for the comment. We agree that the focus is on providing accurate and computationally faster means to predict the optical properties of fractal BC. The following sentence was added to the Abstract:

*In this work, we demonstrate that using a benchmark machine learning algorithm it is possible to make highly fast and accurate predictions of the optical properties for fractal BC.*

**MC 4:** L10: Change "any desired physicochemical properties" to "any desired physicochemical properties within the range of the training dataset".

**R:** Thank you. The change has been made. The Abstract is modified after the MC 1 to 4 as below

It is necessary to accurately determine the optical properties of highly absorbing black carbon (BC) aerosols to estimate their climate impact. In the past, there has been hesitation about using realistic fractal morphologies when simulating BC optical properties due to the complexity involved in the simulations and the cost of the computations. *In this work, we demonstrate that using a benchmark machine learning algorithm, it is possible to make highly fast and accurate predictions of the optical properties for BC fractal aggregates. The mean absolute errors (MAE) for the optical efficiencies ranged between 0.0019 and 0.0039, whereas they ranged between 0.0038 and 0.0044 for the asymmetry parameter.* Unlike the computationally intensive simulations of complex scattering models, the ML-based approach accurately predicts optical properties in a fraction of a second. *Physiochemical properties of BC, such as total particle size (number of primary particles ($N_{pp}$), outer volume equivalent radius ($r_o$), and mobility diameter ($D_m$), outer primary particle size ($a_o$), fractal dimension ($D_f$), wavelength ($l$), the fraction of coating ($f_{coating}$) were used as input parameters for the developed ML-algorithm.* An extensive evaluation procedure was carried out in this study while training the ML algorithms. *The ML-based algorithm compared well with laboratory measurements, demonstrating how realistic morphologies of BC can improve their optical properties. Predictions of optical properties like single scattering albedo ($w$) and mass absorption cross-section (MAC) were improved compared to the conventional Mie-based predictions.* The results indicate that it is possible to generate optical properties in the visible spectrum using BC fractal aggregates with *any desired physicochemical properties within the range of the training dataset*, such as size, morphology, or organic coating. Based on these findings, climate models can improve their radiative forcing estimates using such comprehensive parameterizations for the optical properties of BC based on their aging stages.

**MC 5:** L32: Consider adding an estimate to quantify "good accuracy".

**R:** Thank you for the comment. The MSTM is more commonly used because of its high computational speed and accuracy compared to other numerically exact methods (Kahnert and Kanngießer, 2020). The benefit of using the other popular approach, discrete dipole approximation DDA is that it has no restrictions on the particle shape like the MSTM has. However, it is slower than MSTM. Further, the non-analytical orientation averaging and the choice of dipole spacing in DDA can affect the accuracy of the results (Yurkin and Kahnert, 2013).

*The Multi Sphere T-Matrix (MSTM) method has found widespread applications in the research field because of its high computational speed and accuracy compared to other methods like the DDA (Kahnert and Kanngießer, 2020; Yurkin and Kahnert, 2013).*

*Kahnert, M. and Kanngießer, F.: Modelling optical properties of atmospheric black carbon aerosols, Journal of Quantitative Spectroscopy and Radiative Transfer, 244, 106 849, https://doi.org/10.1016/j.jqsrt.2020.106849, 2020*

*Yurkin, M. A. and Kahnert, M.: Light scattering by a cube: Accuracy limits of the discrete dipole approximation and the T-matrix method, Journal of Quantitative Spectroscopy and Radiative Transfer, 123, 176–183, https://doi.org/https://doi.org/10.1016/j.jqsrt.2012.10.001, 2013*

**MC 6:** L39: Correct the reference of Lamb and Gentine to https://doi.org/10.1038/s41598-023-45235-8

**R:** Thank you for the correction. It has been updated to Lamb and Gentine, 2023.

*Lamb, K. D. and Gentine, P.: Zero-shot learning of aerosol optical properties with graph neural networks, Scientific Reports, 13, 18 777, https://doi.org/10.1038/s41598-023-45235-8, 2023.*

**MC 7:** L43: Please specify the following details regarding the previous machine learning models: the optical properties predicted, the input parameters utilized, the metric employed for model evaluation, the method used to test the model's extrapolation ability, whether the results were compared with actual measurements, and the size of the training dataset. (See major comment 1).

**R:** Thank you for the comment. All the concerns were addressed in the response to major comment 1.

**MC 8:** L68: Please explain the rationale for reporting 35 parameters, even if they are not independent.

**R:** Thank you for the comment. The criteria for selecting 35 parameters have been added to the revised manuscript as:

*The database for the physicochemical and optical properties of BC fractal aggregates has been designed to consider all the possible aging stages of BC. The optical properties of BC fractal aggregates are most sensitive to the change in particle size as they age (Matsui et al., 2018). The particle size is reported as dependent parameters of the number of primary particles ($N_{pp}$), volume equivalent radii ($r_i$ and $r_o$), and mobility diameter ($D_m$). Further, the chemical composition and morphology also influence their optical properties. There are constants related to the particle's chemical composition, such as density and refractive index. The optical properties have been reported as efficiencies and cross-sections. Further dependent optical properties have also been included. The mass and volume of the BC particles were used for conversion between various optical parameters. Further, some parameters, such as the wavelength, were related to the optical model. The database was created using 6192 particles of varying sizes, morphologies, and coating fractions. There are 35 features in the database, which are categorized into 15 physicochemical features, 13 optical features, and seven constants.*

**MC 9:** L69: In Table A1, add a column detailing the step size for the independent parameters.

**R:** Thank you for the comment. A column for values or ranges of the independent parameters has been added to Table A1.

**Table A1.** Features from the database of physicochemical and optical properties of black carbon fractal aggregates. For independent features, the list of values are provided. The features for which the range has provided correspond to dependent features.

| Parameter | Values/Range |
| --- | --- |
| Wavelength ($\lambda$) | 467, 530, 660 |
| Fractal dimension ($D_f$) | 1.5, 1.7, 1.9, 2.1, 2.3, 2.5, 2.7, 2.9 |
| Fraction of coating ($f_{coating}$) | 0, 1, 5, 10, 15, 20, 25, 30, 40, 50, 60, 70, 80, 90 |
| Primary particle size ($a_o$) | 15.1 - 29 |
| Number of primary particles ($N_{pp}$) | 1, 2, 3, 4, 5, 6, 7, 8, 9, 10, 12, 14, 16, 18, 20, 23, 26, 29, 31, 34, 36, 39, 42, 45, 50, 55, 60, 65, 70, 75, 85, 95, 105, 115, 125, 140, 155, 170, 185, 200, 225, 250, 275, 300, 350, 400, 450, 500, 550, 600, 650, 700, 800, 900, 1000 |
| Outer volume equivalent radius ($r_o$) | 12 - 290 |
| Inner volume equivalent radius ($r_i$) | 12 - 150 |
| Mobility diameter ($D_m$) | 17 - 1561 |
| Extinction cross-section ($C_{ext}$) | 0.043 - 3.02 |
| Absorption cross-section ($C_{abs}$) | 0.041 - 1.75 |
| Scattering cross-section ($C_{sca}$) | 0.00038 - 1.82 |
| Asymmetry parameter ($g$) | 0.00036 - 0.91 |
| Single scattering albedo (SSA) | 0.00030 - 0.776 |
| Mass absorption cross-section (MAC) | 3.89 - 24.5 |

**MC 10:** Section 2.1: It may be more appropriate to move this section to the introduction for better flow and context.

**R:** Thank you for the comment. A part of the section has been moved to the introduction. The remaining relevant part has been added to the section 2.1.2 Mixing state.

**MC 11:** Sections 2.2.1 size and morphology, 2.2.2 mixing state, and 2.2.3: I suggest re-arranging these sections. For example: (i) the fractal dimension (and morphology in general) is also related to size, and not only to mixing state; (ii) the wavelength is not an "optical property" of the BC particles (please change also the label in Figure 1).

**R:** Thank you for the comment. The fractal dimension and fraction of coating have been moved to the sub-section of 2.1.2 Mixing state. Wavelength has also been moved to a new sub-section of 2.1.3 Others. The label of Fig. 1 has also been changed as below:

[Figure]

**MC 12:** L74: Replace "formation" with atmospheric aging or processing.

**R:** Thank you for the correction. The word "formation" has been changed to "processing".

**MC 13:** L103: Please specify the step size used for varying the outer radius of the primary particle.

**R:** The outer radius of the primary particle varied according to the fraction of coating. With change of fraction of coating from 0 to 90%, the outer radius of the primary particle changed as 15, 15.1, 15.3, 15.5, 15.8, 16.2, 16.5, 16.9, 17.8, 18.9, 20.4, 22.4, 25.6, and 29. This information has been added to the main text as:

*In contrast, the outer radius of the primary particle ($a_o$), consisting of the organics, varied between 15.1 nm to 30 nm with the fraction of coating ($f_{coating}$) following (Eq. A3) in the Appendix section. The $a_o$ was 15, 15.1, 15.3, 15.5, 15.8, 16.2, 16.5, 16.9, 17.8, 18.9, 20.4, 22.4, 25.6, and 29 according to the value the $f_{coating}$ given in Table A1.*

**MC 14:** L112: Please confirm if you intend "mobility size spectrometers".

**R:** Thank you for the correction. The change has been made.

**MC 15:** L113: Update the citation from Sorensen (2001) to the correct Sorensen (2011) reference:

**R:** Thank you for the correction. The reference has been changed.

**MC 16:** L117: Add "in the atmosphere" to "increasing residence time".

**R:** Thank you for the comment. The change has been made as:

*With increasing residence time in the atmosphere, aggregates become more compact with a fractal dimension of up to 2.2 (Wang et al., 2017).*

**MC 17:** L124: Add the equation for the coating fraction in Appendix A1 for clarity.

**R:** Thank you for the comment. The following equation for the coating fraction has been added in Appendix A1:

*The relationship between the outer radius of the primary particle ($a_o$), the inner radius of the primary particle ($a_i$), and the fraction of organics ($f_{organics}$) is given as:*

$$a_o{}^3 = (1 - f_{organics})a_i{}^3$$

**MC 18:** L141: Discuss in more detail the limitations of the MSTM model, especially how the coated-BC are simulated and how this relates to atmospheric BC.

**R:** Thank you for the comment. The limitations of the MSTM optical model and the coating model are discussed in the revised manuscript:

*L 148: The tunable Diffusion Limited Aggregation (DLA) software (Wozniak et al., 2012) simulated bare BC fractal aggregates of various physicochemical properties. BC can exhibit a range of coating thicknesses and fractal dimensions at any point in the atmosphere, as evidenced by images from Transmission Electron Microscopy (TEM) analyzed from different locations (Fu et al., 2012). Detailed information and images from TEM analysis of BC particles have been provided in the Supplementary. The coating model used in this study is called the "closed-cell model," and the results showed good comparability with the realistic coating model (Kahnert, 2017). The MSTM calculates the electromagnetic properties of a system that consists of a set of spheres (Mishchenko et al., 2004; Mackowski and Mishchenko, 2011).*

*L 158: It was necessary to use this closed-cell coating model due to the non-overlapping sphere limitation of the MSTM code. A sophisticated coating model would be a good choice, but it requires more complex scattering models, such as Discrete Dipole Approximation (DDA), which is computationally expensive.*

**MC 18:** L147: Discuss the relevance of the geometric cross-section to the optical model or consider moving it to the "size" or "Morphology" section for relevance.

**R:** Thank you for the comment. We agree with the reviewer and moved the geometric cross-section to Section 2.1.1 Size.

**MC 19:** L172: As already pointed out, the choice of the ML input parameters should be described more meticulously. Future users of your models should know which input parameters are needed and their validity range. For example, is the wavelength a free parameter, or can it only be chosen among the 3 values used during the training?

**R:** Thank you for the comment. The choice of the ML input parameters has been discussed in more detail:

*The subset of the database used as input was designed to include the critical parameters that influence the BC optical properties. As mentioned in Section 2.1, not all physical properties in the database are independent, as some can be derived from others using simple formulae. Including all properties as inputs for the ML model will thus present it with redundant information, increasing its computational overhead and possibly even harming its performance. The first criterion to narrow down the input parameters was broadly choosing the independent physicochemical parameters representing particle size and mixing state. The fractal dimension ($D_f$) was used to represent the morphology of the BC fractal particles. The chemical mixing state is represented by the fraction of coating ($f_{coating}$). The Wavelength ($\lambda$) is also an input. There was an exception in selecting the input parameters for particle size where we decided to keep four dependent parameters of outer primary particle size ($a_o$), number of primary particles ($N_{pp}$),*

*outer volume equivalent radii ($r_o$), and mobility diameter ($D_m$). The reason for including all four size parameters is that depending upon the focus of a study, the user may have one or another parameter representing the size. In this way, we could provide a more user-friendly prediction script in which the user has a choice to enter at least one or more of the four size parameters. Therefore, the subset of the database's properties as input for our ML models is $\lambda$, $D_f$, $f_{coating}$, $a_o$, $N_{pp}$, and $r_o$, and $D_m$. The range of each input parameter used for designing the prediction algorithm is summarised in Table A1. The ML-algorithm is able accurately predict the optical properties for all input values close or within the ranges provided in Table A1. The selection of input parameters needed while running the prediction script: $\lambda$, $D_f$, $f_{coating}$, and at least one among the $N_{pp}$, $r_o$, and $D_m$.*

The extrapolation and validity range has been discussed in the main text too, please refer to MC 38.

**MC 20:** Does the input parameter "primary particle size" refer to both $a_o$ and $a_i$?

**R:** The input parameter "primary particle size" refers to $a_o$. This has been made checked and made consistent throughout the revised manuscript.

**MC 21:** L176: include the equation describing "g" in Appendix A1 for reference.

**R:** Thank you for the comment. The equation describing the asymmetry parameter is added in the Appendix as:

*The asymmetry parameter (or asymmetry factor) g is defined as the average cosine of the scattering angle theta $\theta$:*

$$g = \langle cos\theta \rangle$$

**MC 22:** L178: Clarify the use of lambda or consider removing it if it is causing confusion with the Box-Cox transformation equation.

**R:** To avoid any confusion with the wavelength, we no longer refer to the transformation parameter as lambda in the revised manuscript.

**MC 23:** L187: Specify in the text that "Fro" stands for Frobenius norm.

**R:** We added a definition of the Frobenius norm after Eq. (2).

**MC 24:** L191: Mention if the polynomial kernel was part of the experimentation and provide insights or remove (it is difficult to clearly identify which components you tried or not, and which ones resulted in the best model).

**R:** Indeed, the polynomial kernel was used in early experimentation but did not yield any promising results so we dropped it before running the final suite of experiments. To clear things up, we no longer mention the polynomial kernel as an example.

**MC 25:** L193: Specify that it is the L2-norm in the main text for clarity.

**R:** We added a short definition analogue to the L1 norm after Eq. (4).

**MC 26:** L232: Please clarify the distribution of data among training, testing, and validation sets for the three experiments. Specifically, what percentage of the data is excluded from training and used for validation in the interpolation and extrapolation experiments? It is unclear whether the extrapolation was performed on one or two sets:

Table 1 suggests that the model was trained once using the combined ranges for Df, while the caption of Figure 3 indicates a Df training range of [1.5, 2.5). Conversely, Table B4 implies that the model underwent training twice, each time with a different Df range.

**R:** For each training/test split mentioned in the text, we randomly choose 30% of the training split as a held-out validation set. We trained a model for each of the interpolation/extrapolation splits mentioned in Tables B3 and B4, adding up to a total of 6 models, not including the model trained on the random split. We adapted our explanations in the main text to make this clearer for the reader.

**MC 27:** L236: Discuss if the data was divided into batches during training to improve generalization.

**R:** We are not entirely sure if we understand the comment correctly. Our neural network training was indeed performed using a variant of mini-batch stochastic gradient descent, but we do this due to memory and time constraints, not necessarily to improve generalization. If we misunderstood your comment, we would kindly ask you to provide clarification on this matter.

**MC 28:** L238: Briefly mention the ranges of the parameters used or excluded from the training also in the main text (in addition to Table B3 and B4).

**R:** The range of the training data used for the two splits of interpolation, and four splits of extrapolation have been briefly mentioned in the main text.

**MC 29:** L243: Consider reporting (in the Appendix would be sufficient) the Mean Absolute Percentage Error (MAPE) in addition to the Mean Absolute Error (MAE), as MAPE provides a more sensitive measure compared to the relative percentage error derived from MAE.

**R:** We decided against using the MAPE when reporting our results as it comes with certain characteristics that make the MAE, in our eyes, the more appropriate choice for our use case. More specifically, consider a data point where the true value is 0.1 and the prediction is 0.2 and another point where the true value is 1.8 and the prediction 1.9. The absolute error is the same in both cases (0.1), but the relative error is quite different, i.e., 100% and ~5.6%, respectively. In our view, the prediction error should be weighted equally for both points, and therefore, we chose the MAE as our error metric.

For reference, error distributions for the ML methods shown are presented in terms of MAPE in the Supplementary.

[Figure]

*Figure S1. Boxplots summarizing the mean absolute percentage error (MAPE) between the predicted value and the true value for three optical properties with respect to $D_f$.*

[Figure]

*Figure S2. Boxplots summarizing the mean absolute percentage error (MAPE) between the predicted value and the true value for three optical properties with respect to $f_{coating}$.*

[Figure]

*Figure S3. Boxplots summarizing the mean absolute percentage error (MAPE) between the predicted value and the true value for three optical properties with respect to $D_m$.*

**MC 30:** Figure 3: Could you provide a similar boxplot for the Mean Absolute Percentage Error (MAPE)? Additionally, please include a comprehensive description of the boxplot's features. Also, consider reassessing the inclusion and visual representation of outliers in the plots.

**R:** Please refer to the response of MC 29 for the figures of the boxplot for the Mean Absolute Percentage Error (MAPE).

The following description of the box plot has been added to the caption of each boxplot-containing figure:

*The lower hinge and the upper hinge of the boxplot represent the 25 % and 75 % quantile of the observations, respectively....*

Including the outliers significantly reduced the visualization of the boxplots and, therefore, were omitted from the figures. However, please note that all the outliers are considered in the training data and error evaluation.

**MC 31:** Figure 3: How would it look like for the MAPE? Please include a detailed description of the features in the boxplot, such as the representation of the whiskers, median, quartiles, etc. Reassess the inclusion of outliers in plots, their number and distribution are important to evaluate the model performance (e.g., what is the probability that the prediction is an outlier?).

**R:** Please refer to the response of MC 29 for the figures of the boxplot for the Mean Absolute Percentage Error (MAPE). The description of the box plot has been added to all the boxplots, as specified in MC 30. Including the outliers significantly reduced the visualization of the boxplots and, therefore, were omitted from the figures. However, please note that all the outliers are considered in the training data and error evaluation.

**MC 32:** Figure 4: Add the blue line to the legend for clarity.

**R:** The description of the blue line has been added to the figure caption as:

*Figure 4. Comparison of the predicted optical properties with their true values when the ML models are trained on a random subset of data. The data points for predicted optical properties correspond to KRR and ANN, as shown by the legend on the top right. The blue line in each panel of the figure corresponds to the one-to-one line between the X-axis and Y-axis.*

**MC 33:** Figure 5: Why don't you plot MAE or MAPE here?

**R:** Please refer to the response of MC 29 for the reason why we used MAE instead of MAPE.

**MC 34:** Section 5: Consider changing the section's title. This is a comparison to laboratory measurements and not "Atmospheric implications".

**R:** Thank you for the comment. The Section's title has been changed to "Comparison to black carbon laboratory measurements".

**MC 35:** Figure 6: Please include a legend for the black line and shaded area. Additionally, consider evaluating and reporting the errors associated with the measured Single Scattering Albedo and incorporate error bars (x-axis) in the plot. Can you add the exact results of the MSTM simulations obtained with the same parameters used for the ML model?

**R:** The suggestions made by the reviewer have been included in the revised figure as shown below:

[Figure]

*Figure 6. Single scattering albedo ω for coated BC particles generated in a laboratory study at different $f_{coating}$ (Romshoo et al., 2022). (A) compares the $\omega_{ML}$ with the measured ω from the laboratory experiment. (B) compares the $\omega_{Mie}$ with the measured ω. The ML results correspond to KRR, the default algorithm used in the prediction script. Error bars along the X-axis show the uncertainty in the measured ω. The cross points are the ω from MSTM*

*simulations. The black linear represents a linear regression equation shown in the upper left corner, with the coefficient of determination in the upper right corner of each panel. The grey area represents the 95% confidence level interval for predictions.*

**MC 36:** L310: Specify which ML algorithm is being used at this point in the text for clarity.

**R:** Thank you for the comment. The ML algorithm has been used in the default Kernel Ridge Regression method. This has been made clear in the main text and in Fig. 6.

**MC 37:** Section 5.1: The limited number of measurements used for comparison with the model should be acknowledged as a limitation. Additionally, future work should aim to increase the dataset size for more robust validation.

**R:** Thank you for the comment. The limitations and outlook have been added to the revised manuscript as:

*In this study, due to the experimental design of (Romshoo et al., 2022), we could only test the ML-based prediction algorithm for particles with $f_{coating}$ of less than 65 %. The extension of the current algorithm to include more parameters also demands closure studies using more datasets of laboratory and ambient measurements.*

**MC 38:** L332: What considerations have been made regarding the potential for extrapolating additional parameters? Are there specific constraints on allowable input values? For example, how does the model perform when using more than 1000 primary spheres or a primary particle size different from 15 nm? Demonstrating the model's ability to extrapolate (or not) with these parameters would be advantageous. Alternatively, please specify if the model is designed to accept only those parameters that fall within the training data range, as opposed to 'any reasonable inputs'.

**R:** Thank you for the comment. The following paragraph has been added to discuss the potential for extrapolating additional parameters like particle size, primary particle size, and wavelength:

*In this study, the ML-based prediction algorithm is developed using training data of $N_{pp}$ up to 1000, corresponding to particles with a maximum $D_m$ of 1561 nm depending on the $f_{coating}$. This range of particle sizes was chosen while designing the database, considering the realistic size of BC-containing particles in the atmosphere. TEM analysis has shown a high probability that the BC-containing particles less than 1500 nm will be fractal (Adachi et al., 2016; Wang et al., 2017). The ML algorithm developed in this study, based on a close-shell coating model, is suitable for particles smaller than 1500 nm. However, when aerosol particles grow larger, the mass of BC decreases significantly compared to the mass of the coating (Adachi et al., 2016). For this case, using the conventional core-shell-based spherical morphology is appropriate. This is why we limited our training data range for particle size to 1561 nm. However, as demonstrated by Luo et al. (2018a), adding a few points to the training data significantly improves the extrapolation efficiency of machine learning models. Further, some studies show that the optical properties are not sensitive to the change in the primary particle size a. Therefore, we fixed the $a_i$ to 15 nm, and changed $a_o$ from 15.1 to 29 depending on the $f_{coating}$. Similar to the parameters related to particle size, such as $N_{pp}$, $r_o$, and $D_m$, adding a few data points to the $a_i$ or $a_o$ can help optimize the extrapolation ability of the ML-based prediction algorithm. Although future studies can extend the model's extrapolation ability, the particle size range of the current prediction algorithm covers the physically feasible cases for BC fractal aggregates.*

*The prediction script can predict the optical properties well for the range between 467 and 660, and points close to the upper and lower limit.*

**MC 39:** Table A1: The wavelength shouldn't be presented as a continuous parameter since test at intermediate values are not tested.

**R:** Thank you for the comment. The wavelength has been presented as a discrete parameter in Table A1.

**MC 40:** Figure C2: I couldn't find any reference to this figure in the main text.

**R:** Thank you for the comment. Figure C2 has been referred to in the revised main text as:

*The interpolation and extrapolation results are similar if training and test data are split according to the parameters of the coating $f_{coating}$ and particle size $D_m$. The appendix provides a more detailed discussion about the interpolation and extrapolation results for parameters of coating $f_{coating}$ and particle size $D_m$ in Fig. C1 and Fig. C2, respectively. Overall, the narrow box plots of the errors in the random split demonstrate the effectiveness of the ML algorithms for predicting the optical properties of coated BC fractal aggregates.*

**MC 41:** Figure C7: Does it contain the same number of measurements as presented in Figure 6?

**R:** Figure C7 contains fewer measurements since the mass measurements from TEOM used to calculate the observational MAC were available only during coated experiment 2. The following information has been added to the Appendix: Laboratory measurements of black carbon.

*The predicted MAC is compared to six experimental cases of coated soot for which the observational MAC was available (last six rows in Table 1 in Romshoo et al., 2022).*

---

## Author Comment (AC2)

**Comment 1**: Romshoo et al. (egusphere-2023-2400) simulated bare soot aggregates by diffusion-limited cluster aggregation (DLCA) and coated soot by adding spherical coatings around the spherical monomers of the DLCA aggregates. The authors then applied machine-learning models to interpolate numerically accurate MSTM calculations of the aggregates' optical properties.

The approach taken by the present authors was taken by two different studies previously. Luo et al. (2018) considered soot aggregates but not coatings. Lamb and Gentine (2021) considered uncoated and coated soot, using a coating model almost identical to the present authors. The authors erroneously wrote that Lamb and Gentine do not consider coating", but the only difference in their approach is an insignificant change in the machine-learning model. For this reason, I have to recommend rejection of the present manuscript.

**Response**: Thank you for your comment. According to our understanding, Lamb and Gentine (2021) generated the optical properties of bare BC fractal aggregates using a graph neural network (GNN). We found strong evidence that they did take coating into account. On page 2 of the paper is written:

"Here, we show the optical properties of bare BC with complex morphology can be accurately predicted with a graph neural network (GNN) by representing BC fractal aggregates as networks of interacting spheres. GNNs are recently developed machine learning algorithms that learn on graph-structured data sets, allowing models to include arbitrary relational information directly."

Furthermore, on Page 8, the authors wrote, "As a proof of concept, we have trained a GNN to predict the optical properties of bare BC fractal aggregates with a range of different fractal parameters."

On page 8 of the paper, **as an outlook**, it is written that "The GNN approach provides an obvious extension to internally mixed aerosols (Fig. 1), as the thickness of coatings and their indices of refraction or organic fraction could be included as additional node-level features (in the thinly coated case) or graph-level features (for the thickly coated case)."

The above sentence could have led to an easy misinterpretation, but we checked that there is no indication of coating in Table S1 of the parameters provided in the supplementary material. They also used refractive indices with an imaginary part higher than 0.4, which is typical for bare BC aggregates. The authors of Lamb and Gentine (2021) mean that others can extend their approach to include other node features like coating. Therefore, we respectfully disagree with your comment.

Please find the recently published version of Lamb and Gentine (2021) below:

https://www.nature.com/articles/s41598-023-45235-8

**Comment 2**: Independently, I also cannot recommend publication of this manuscript as the coating model is completely unrealistic. No experiment has ever observed coated soot to retain its original shape while adding spherical coatings to the monomers. Romshoo et al., and several others have already published this model, but it contradicts dozens of smog chamber and field studies using electron microscopy, which all observed restructuring. There is no value in using machine-learning algorithms to interpolate the results of an inaccurate model.

**Response**: Thank you for your comment. Atmospheric soot undergoes various processes, including a possible restructuring after emission, depending on multiple factors such as geographical location, atmospheric chemistry, and meteorology (Fig. 1 adapted from Sedlacek et al., 2022). The lifecycle of soot particles is not captured properly by global climate models. Current state of art for representing atmospheric soot particles focusses on spherical aged particles. At any point in the atmosphere, BC can exhibit a wide range of morphologies showing diversity at different locations (Fig. 2 taken from Fu et al., 2012). It was observed that aged transported soot can retain its fractal morphology 500 to 1000 km downwind of emission sources (Fig. 3 adapted from Sun et al., 2020). The model provided in this study was designed to simulate the optical properties for the entire BC lifecycle capturing the transition between fresh fractal to aged spherical particles.

[Figure]

Figure 1. Lifecycle of BC particles adapted from Sedlacek et al., 2022.

[Figure]

Figure 2. Transmission electron microscopy (TEM) images adapted from Fu et al., 2012 showing wide range of BC morphologies at different locations.

[Figure]

Figure 3. Results from TEM analysis showing that aged transported soot can retain its fractal morphology 500 to 1000 km downwind of emission source, taken from Sun et al., 2022.

The restructured soot is most representative of the "embedded" soot. However, BC particles can have thin, medium, or partial coatings as well. For this reason, we provide a ML algorithm that provides optical properties for all possible ranges of fractal dimensions between 1.5 and 2.9. Furthermore, it depends on the user to choose a suitable fractal dimension and fraction of coating to represent the BC particles they want to simulate. For example, for restructured aged particles, the user can choose a fractal dimension higher than 2.5, and coating fractions higher than 50 %.

The coating model used in this study is called the "closed-cell model," the results showed good comparability with the realistic coating model (Kahnert 2017). It was necessary to use this coating model due to the limitations of the MSTM code that was used for generating the optical properties. We agree that a more sophisticated coating model would be a good choice, but it requires more complex scattering models, such as Discrete Dipole Approximation (DDA), which is computationally expensive. With the DDA method, generating elaborate datasets for training ML algorithms is not feasible.

Simulations of BC's optical properties are required for global climate models. Presently, the simplistic Mie core-shell model is used for BC particles, representing the aged portion of the BC lifecycle. Although a more realistic coating model would be ideal, our ML method offers a robust solution for predicting the present scenario due to the above-discussed limitations. We provide a method that predicts the optical properties of a wide range of ambient soot particles with high accuracy. Using this method, we can overcome the limitations of the simplistic core-shell model, which only represents aged BC particles. Furthermore, calibration of light absorption measurement devices mostly is done with fresh soot. We can make the link to atmospheric relevant absorption by simulating mass absorption cross-sections and light absorption enhancement factors.

Therefore, the results of this study are valuable for the simulation of realistic scenarios, despite the model limitations. We acknowledge that there is scope for future studies to extend such an ML-based approach using other morphological models of BC and coating positions. Reviewer's feedback will be carefully considered in the revised manuscript, providing a detailed explanation of our approach and its limitations in the revised manuscript.

**References:**

Fu, H., Zhang, M., Li, W., Chen, J., Wang, L., Quan, X., and Wang, W.: Morphology, composition and mixing state of individual carbonaceous aerosol in urban Shanghai, Atmos. Chem. Phys., 12, 693–707, https://doi.org/10.5194/acp-12-693-2012, 2012.

Sun, C., Adachi, K., Misawa, K., Cheung, H. C., Chou, C. C. K., & Takegawa, N. (2020). Mixing State of Black Carbon Particles in Asian Outflow Observed at a Remote Site in Taiwan in the Spring of 2017. Journal of Geophysical Research: Atmospheres, 125(16). https://doi.org/10.1029/2020JD032526

Kahnert, M.: Optical properties of black carbon aerosols encapsulated in a shell of sulfate: comparison of the closed cell model with a coated aggregate model, Opt. Express, 25, 24579–24593 https://doi.org/10.1364/oe.25.024579, 2017.

Sedlacek, A. J., Lewis, E. R., Onasch, T. B., Zuidema, P., Redemann, J., Jaffe, D., & Kleinman, L. I. (2022). Using the Black Carbon Particle Mixing State to Characterize the Lifecycle of Biomass Burning Aerosols. Cite This: Environ. Sci. Technol, 2022, 14315–14325. https://doi.org/10.1021/acs.est.2c03851

---

## Author Response (AR2)

Dear Editor and Reviewer, we would like to thank you for your positive response and acknowledgement. In response to the Reviewer's suggestions, we have revised the manuscript in the following manner:

**Comment 1:** Specifically, add a comment that Lamb and Gentine did not model coated aggregates. This point was not emphasized clearly enough by Lamb and Gentine and it should be emphasized here in Line 55.

**Response**: Thank you for the comment. In the revised version, we have specified that Lamb and Gentine did not model coated aggregates.

**Comment 2:** The use of a miniCAST at different setpoints is not representative of the coated atmospheric soot particles the authors cited in their rebuttal. However, it is reasonable to use these particles for validation of the present work. Please clarify this in the abstract ("laboratory measurements of miniCAST soot") and Figure 6 caption ("laboratory study at different miniCAST setpoints" instead of "at different forganics").

**Response:** Thank you for the comment. In the revised version, we have specified that measurements were laboratory-based from a miniCAST soot generator.